
# Evaluation of asymmetric Oxygen Minimum Zones in the tropical Pacific: a basin-scale OGCM-DMEC V1.0

Kai Wang[1], Xiujun Wang[1,2*], Raghu Murtugudde[2], Dongxiao Zhang[3], Rong-Hua Zhang[4]

[1]College of Global Change and Earth System Science, Beijing Normal University, Beijing 100875, China
[2]Earth System Science Interdisciplinary Center, University of Maryland, College Park, Maryland 20740, USA
[3]JISAO, University of Washington and NOAA, Pacific Marine Environmental Laboratory, Seattle, Washington 98115, USA
[4]Institute of Oceanology, Chinese Academy of Sciences, Qingdao, Shandong 266071, China

*Correspondence to*: Xiujun Wang (xwang@bnu.edu.cn)

**Abstract.** The tropical Pacific Ocean holds the world's two largest Oxygen Minimum Zones (OMZs), showing a prominent
hemispheric asymmetry, with a much stronger and broader OMZ north of the equator. However, there is a lack of
quantitative assessments of physical and biological regulations on the asymmetry of tropical Pacific OMZs. Here, we apply a
fully coupled basin-scale model (OGCM-DMEC V1.0) to investigate the impacts of physical supply and biological
consumption on the dynamics of OMZs in the tropical Pacific. We first utilize observational data to evaluate and improve
our model simulation, and find that mid-depth DO is more sensitive to the parameterization of background diffusion.
Enhanced background diffusion results in higher DO concentrations at mid-depth, leading to significant improvement of our
model capability to reproduce the asymmetric OMZs. Our study shows that while physical supply of DO is increased in
majority of the tropical Pacific due to enhanced background diffusion, there is little increase in the largest OMZ to the north.
Interestingly, enhanced background diffusion results in lower rates of biological consumption over ~300-1000 m in the entire
basin, which is associated with redistribution of dissolved organic matter (DOM). Our analyses demonstrate that weaker
physical supply in the ETNP is the dominant process responsible for the asymmetric DO in the core OMZs (~200-600 m)
while higher biological consumption to the north plays a larger role in regulating DO concentration beneath the OMZs
(~600-800 m), with implication for the asymmetric OMZs. This study highlights the roles of physical supply and biological
consumption in shaping the asymmetric OMZs in the tropical Pacific, underscoring the need to understand both physical and
biological processes for accurate projections of DO variability.

## 1 Introduction

Photosynthesis and respiration are important processes in all ecosystems on the Earth, with carbon and oxygen being the two
main elements. The carbon cycle has garnered much attentions and made significant process in both the observations and
modelling of biological processes (e.g., uptake of $CO_2$ and respiration), and physical/chemical processes (e.g., the fluxes
between the atmosphere, land and ocean). However, the oxygen cycle has received much less attention despite its large role
in the earth system (Breitburg et al., 2018; Oschlies et al., 2018).



Dissolved oxygen (DO) is a sensitive indicator of physical and biogeochemical processes in the ocean thus a key parameter for understanding the ocean's role in the climate system (Stramma et al., 2010). In addition to photosynthesis and respiration, the distribution of DO in the world's oceans is also regulated by air-sea gas exchange, ocean circulation and ventilation

(Bopp et al., 2002; Brandt et al., 2015; Levin, 2018). Unlike most dissolved nutrients that display an increase in concentration with depth, DO concentration is generally low at mid-depth of the ocean. The most remarkable feature in the oceanic oxygen dynamics is the so-called Oxygen Minimum Zone (OMZ) that is often present below 200 m in the open oceans (Karstensen et al., 2008; Stramma et al., 2008), where DO concentration is less than 20 mmol m$^{-3}$ (Gilly et al., 2013; Paulmier and Ruiz-Pino, 2009).


The world's two largest OMZs are observed at ~200-700 m in the Eastern Tropical North Pacific (ETNP) and South Pacific (ETSP), showing a peculiar asymmetric structure across the equator, i.e., a much larger volume of suboxic water (<20 mmol m$^{-3}$) to the north than to the south (Karstensen et al., 2008; Paulmier and Ruiz-Pino, 2009; Stramma et al., 2008). It is known that OMZs are caused by the biological consumption associated with remineralization of organic matter (OM), and weak

physical supply of DO due to sluggish subsurface ocean circulation and ventilation (Brandt et al., 2015; Kalvelage et al., 2015). Although there have been a number of observation based analyses addressing the dynamics of OMZs in the tropical Pacific during the past decade (Czeschel et al., 2012; Garçon et al., 2019; Schmidtko et al., 2017; Stramma et al., 2010), our understanding is uncompleted in terms of the underlying mechanisms that regulate DO dynamics at mid-depth due to the limitation of available data (Oschlies et al., 2018; Stramma et al., 2012).


Large-scale physical-biogeochemical models have become a useful tool for advance understanding of the oceanic oxygen cycle (Cabré et al., 2015; Ward et al., 2018; Williams et al., 2014), and have demonstrated that physical processes such as ventilation and circulation play a dominant role in regulating the dynamics of tropical OMZs (Deutsch et al., 2006; Ito et al., 2019; Montes et al., 2014; Stramma et al., 2010). However, many current global and regional models could not consistently

reproduce the dynamics of tropical Pacific OMZs (Cabré et al., 2015; Stramma et al., 2012), which may be related to unresolved transport process, unaccounted biological consumption or missing biogeochemical feedbacks in the models (Keller et al., 2016; Palter and Trossman, 2018; Shigemitsu et al., 2017). Moreover, there has been a lack of quantitative assessments on the relative roles of physical supply and biological consumption in driving the tropical asymmetric OMZs using validated models (Keller et al., 2016; Oschlies et al., 2018).


A fully coupled basin-scale physical-biogeochemical model (OGCM-DMEC V1.0) was developed for the tropical Pacific (Wang et al., 2008; Wang et al., 2015; Wang et al., 2009), which shows capability of reproducing observed spatial and temporal variations in physical, nutrient and carbon fields. In this study, we use the validated basin-scale model to evaluate the simulation of oxygen cycle, focusing on the dynamics of mid-depth DO. We first carry out model evaluation and



validation by using observational DO data to improve the simulation of OMZs in the tropical Pacific. Then, we use the improved model to quantify how physical supply and biological consumption regulate the dynamics of DO at mid-depth. The objective of this study is to advance our model capacity to simulate the oceanic oxygen cycle, and to explore the mechanisms driving the asymmetric OMZs in the tropical Pacific.

## 2. Model description

The OGCM-DMEC V1.0 is a fully coupled basin-scale physical-biogeochemical model that has shown a good model-data agreement in the carbon cycle for the tropical Pacific Ocean (Wang et al., 2015). A brief description is presented in this section.

### 2.1 Ocean physical model

The basin-scale OGCM, a reduced-gravity, primitive-equation, sigma-coordinate model, is coupled to an advective atmospheric model (Murtugudde et al., 1996). There are 20 layers with variable thicknesses in the OGCM. The mixed layer (the upper-most layer) is determined by the Chen mixing scheme (Chen et al., 1994), which varies from 10 m to 50 m on the equator. The remaining layers in the euphotic zone are approximately 10 m in thickness. The model domain is between 30ºS and 30ºN, and zonal resolution is 1º. Meridional resolution varies between 0.3º and 0.6º over 15ºS-15ºN (1/3° over 10ºªS-
10ºN), and increases to 2º in the southern and northern "sponge layers" (the 20º-30º bands) where temperature, salinity, nitrate and iron are gradually relaxed back towards the observed climatological seasonal means from the World Ocean Atlas, 2013 (WOA2013: http://www.nodc.noaa.gov/OC5/woa13/pubwoa13.html).

The model is forced by atmospheric conditions: climatological monthly means of solar radiation and cloudiness, and
interannual, 6-day means of precipitation and surface wind stress. Precipitation is from ftp://ftp.cdc.noaa.gov/Datasets/gpcp, and surface wind stresses from the National Centers for Environmental Prediction (NCEP) reanalysis (Kalnay et al., 1996). Air temperature and humidity above the ocean surface are computed by the atmospheric mixed layer model. Initial conditions were obtained from outputs of an interannual hindcast simulation over 1948-1978, which itself is initialized from a climatological run with a 30-year spin up. The initial conditions for the climatological spin up are specified from the
WOA2013. We carry out an interannual simulation for the period of 1978-2000, and analyze model output for the period of 1981-2000.


## 2.2 Ocean biogeochemical model

The dynamic marine ecosystem-carbon (DMEC) model is the main part of the biogeochemical model that is embedded in the

basin-scale OGCM. The DMEC model consists of eleven components: small (S) and large (L) sizes of phytoplankton ($P_S$ and $P_L$), zooplankton ($Z_S$ and $Z_L$) and detritus ($D_S$ and $D_L$), DON, ammonium, nitrate, dissolved iron, and DO (Figure 1). All biological components use nitrogen as their unit, and are computed in a manner similar to physical variables.

In this model, net community production (NCP) is computed as:

$$NCP = 6.625 * (\mu_s P_s + \mu_L P_L - r_s Z_s - r_L Z_L - c_{DON} DON - c_{DS} D_s - c_{DL} D_L) \tag{1}$$

where 6.625 is the C:N ratio, $\mu$ the rate of phytoplankton growth, $r$ the rate of zooplankton respiration, $c$ the rates of detritus decomposition and DON remineralization. The equations for biogeochemical processes and model parameters have been reported in details previously (Wang et al., 2008; Wang et al., 2015).

## 2.3 Computation of oxygen sources and sinks

The time evolution of DO is regulated by physical, biological and chemical processes:

$$\frac{\partial O_2}{\partial t} = -u \frac{\partial O_2}{\partial x} - v \frac{\partial O_2}{\partial y} - w \frac{\partial O_2}{\partial z} + O_{mix} + O_{bio} + O_{gas} \tag{2}$$

where $u$, $v$, and $w$ are zonal, meridional, and vertical velocity, respectively. $O_{mix}$ is the vertical mixing term that is calculated by three subroutines (Chen et al., 1994). Briefly, the first one computes convection to remove instabilities in the water column, and the second one determines the mixed layer depth. The third one computes partial mixing ($P_m$) between two adjacent layers to relieve gradient Richardson number instability, which is calculated as follows:

$$P_m = \left(1 - \left(\frac{Ri}{0.7}\right)^\lambda\right)(Ri \le 0.7) \tag{3}$$

$$P_m = 0 \ (Ri > 0.7) \tag{4}$$

where the mixing parameter $\lambda$ is set to 1.

The biological source/sink term $O_{bio}$ is computed as follows:

$$O_{bio} = -1.3 * NCP \tag{5}$$

where 1.3 is the O:C Redfield ratio.

The flux of O₂ from the atmosphere to the surface ocean is computed as:

$$O_{gas} = (O_{Sat} - O)K_0 \tag{6}$$

where $O_{sat}$ is the O₂ saturation, a function of temperature and salinity (Weiss, 1970), and $K_0$ the gas transfer velocity that is a function of wind speed ($u_s$) and SST according to Wanninkhof (1992):

$$K_0 = 0.31 u_s^2 \sqrt{\frac{Sc}{Sc20}} \tag{7}$$





where $Sc$ ($Sc_{20}$) is the Schmidt number at SST (at 20° C):

$$S_c = 1953 - 128T + 3.99T^2 - 0.05T^3 \tag{8}$$

**3. Model evaluation and experiments**

**3.1 Evaluation of DO concentrations**

We first evaluate simulated DO for the tropical Pacific Ocean using the outputs from OGCM-DMEC V1.0 (hereafter reference run). We focus on model-data comparisons for the layers of 0-200 m and 200-600 m that broadly represent the euphotic zone and OMZ, respectively. For the 0-200 m, the observed large-scale spatial pattern of DO is captured by the

reference run, both data and model output show the low DO waters (<120 mmol m$^{-3}$) in the eastern tropical Pacific Ocean and supersaturated DO (>200 mmol m$^{-3}$) in majority of other sections (Figure 2a and 2c). Overall, the bias from the reference run is very small (<10 mmol m$^{-3}$) in the surface water (Figure 2e). For the mid-depth, the WOA2013 data shows much larger area of suboxic waters (<20 mmol m$^{-3}$) in the ETNP than in the ETSP (Figure 2b). Although the reference run produces two OMZs off the equator (Figure 2d), the size of suboxic waters in the ETNP is much larger in the reference run than that in the

WOA2013 data. The reference run thus underestimates DO over 200-600 m, particularly in the eastern tropical Pacific, with a bias of 10-20 mmol m$^{-3}$ (Figure 2f). The relative roles of the physics vs. the biogeochemistry in determining the bias are diagnosed further below.

**3.2 Sensitive experiments**

The underestimated DO at mid-depth may be a result of overestimation of removal (primarily the consumption associated with remineralization of organic materials) and/or underestimation of supply (e.g., vertical mixing). We conduct a sensitivity study that consists of four new simulations. The first two experiments are designed to reduce the biological consumption of DO: by reducing DON remineralization constant ($c_{DON}$ in equation 1) by 50% (Exp1), or changing O:C utilization ratio ($R_{OC}$ in equation 5) from 1.3 to 1.0 (Exp2); the other two are to increase supply of DO by changing the partial mixing parameter

($P_m$ in equation 4 ) from 0 to $10^{-4}$ m$^{-2}$ s$^{-1}$ (Exp3), or applying a variable $P_m$ (Table 1).

The comparisons between simulated DO and WOA2013 climatology data are shown in Figure 3 and 4. Figure 3a illustrates the observed vertical-zonal variations of DO over 10°S-15°N, showing small volume of suboxic water (<20 mmol m$^{-3}$) centered at ~400-600 m from 110°W to 90°W. Similar to the reference run (Figure 3b), Exp1 and Exp2 also under-estimates

DO concentration at mid-depth (Figure 3c and 3d), showing much larger volume of suboxic waters than that from WOA2013. On the other hand, Exp3 and Exp4 produce reasonable DO concentrations at mid-depth, particularly in the east





of 120°W (Figure 3e and 3f), although the simulated volume of suboxic water is slightly larger than from the observation data (Figure 3a).

Figure 4a illustrates a larger volume of suboxic water located north of ~4°N and a smaller volume of suboxic water over 8°S-4°S, which are separated by relatively higher DO (>30 mmol m$^{-3}$) along the equator. Both Exp1 and Exp2 (Figure 4c and 4d), as well as the reference run (Figure 4b), produce much larger volumes of suboxic water that are extend to the equatorial region, and even merge into one. Exp3 and Exp4 (Figure 4e and 4f) capture the observed basin-scale spatial distribution of DO, particularly the asymmetric feature (i.e., a larger volume of suboxic water to the north but a smaller size

of suboxic water to the south), and relatively higher DO (~40 mmol m$^{-3}$) over 4°S-2°N. Many ESMs are unable to reproduce the positions and sizes of the OMZs in the eastern tropical Pacific Ocean (Bao and Li, 2016; Cabré et al., 2015); some models overestimated the extent of suboxic water, which might be due to over-estimated productivity in the euphotic zone (Moore et al., 2013; Williams et al., 2014).

## 3.3 Model skill assessment

To further evaluate the performance of experiments on OMZs, four statistical measures are applied over 300-500 m in the ETNP (150°W-90°W, 0°-10°N) and ETSP (150°W-90°W, 10°S-0°). As showed in Table 2, compared to the reference run, both bias and RMSE decrease in all four experiments, with the smallest bias (-8.59 mmol m$^{-3}$ in ETNP and 1.05 mmol m$^{-3}$ in ETSP) and RMSE (12.36 mmol m$^{-3}$ in ETNP and 8.43 mmol m$^{-3}$ in ETSP) found in Exp3. Many current models show a

maximum of RMSE (~20-80 mmol m$^{-3}$) with respect to observed DO from mixed layer to 1000 m (Bao and Li, 2016; Cabré et al., 2015). Table 1 and Figure 5 illustrate that Exp3 produces the best outputs, with the highest correlation coefficient (0.72 in ETNP and 0.893 in ETSP) and also close-to-1 NSD (1.03 in ETNP and 1.71 in ETSP).

## 4 Results and discussions

In this section, we further compare the improved model simulation (EXP3) and reference run to diagnose the relative

contributions of physical supply and biological consumption to the asymmetric OMZs in the tropical Pacific, aiming to identify the underlying mechanisms regulating the dynamics of mid-depth DO.

## 4.1 Distribution of DO and physical supple at mid-depth

We first compare the distribution of DO over 300-500 m between reference run and Exp3. The reference run produces much

large volume of suboxic waters (<20 mmol m$^{-3}$) in both the ETNP and ETSP where the two OMZs are merged (Figure 6a). On the other hand, Exp3 performs well in reproducing the sizes and locations of two asymmetric OMZs (Figure 6b). Mid-



depth DO concentration is 1-20 mmol m$^{-3}$ higher in the Exp3 than in the reference run, with the largest difference in the western equatorial Pacific (Figure 6c). However, despite of the small absolute increase (1-5 mmol m$^{-3}$) in the ETNP, the relative increase of DO is approximately 200-400%, indicating that mid-depth DO in the north OMZ is more sensitive to the
parameterization of physical processes.

The overall increased DO at mid-depth is likely a result of stronger physical supply, as seen below, due to enhanced background diffusion in Exp3. Numerous studies have indicated that physical mixing is the only source of DO for OMZs (Brandt et al., 2015; Czeschel et al., 2012; Talley et al., 2016), particularly, turbulent diffusion accounted for 89% of the net
DO supply for the core OMZ layer (Llanillo et al., 2018). The reference run in our study shows a range of 4-11 mmol m$^{-3}$ yr$^{-1}$ in physical supply over the 300-500 m, with higher rates (>11 mmol m$^{-3}$ yr$^{-1}$) found mainly in the northern central tropical Pacific, which is located inside of the suboxic waters (Figure 7a). Exp3 produces generally higher rates (5-12 mmol m$^{-3}$ yr$^{-1}$) of physical supply, with a larger size of high-rate (>11 mmol m$^{-3}$ yr$^{-1}$) water near the edge of northern suboxic water (Figure 7b). There have been a handful of modelling studies of physical supply rate for the mid-depth waters of tropical Pacific,
which show similar magnitudes (from ~4 to ~8 mmol m$^{-3}$ yr$^{-1}$) (Llanillo et al., 2018; Shigemitsu et al., 2017). Figure 7c illustrates that physical supply is increased by ~0.5-1 mmol m$^{-3}$ yr$^{-1}$ in most of the mid-depth, with the highest increase in the southern part of central equatorial Pacific. Interestingly, there is somehow a small decrease (by ~0.1 mmol m$^{-3}$ yr$^{-1}$) of physical supply in the ETNP-OMZ, implying that changes in other processes such as biological consumption may also contribute to the enhanced DO at mid-depth.


## 4.2 Biological consumption of DO at mid-depth

We further compare biological consumption of DO between Exp3 and the reference run (Figure 8). For the reference run, biological consumption rate is <6 mmol m$^{-3}$ yr$^{-1}$ off the equator and >10 mmol m$^{-3}$ yr$^{-1}$ in most of the upwelling region, with higher rates (>11 mmol m$^{-3}$ yr$^{-1}$) mainly located in the northern central tropical Pacific, and in the suboxic waters (Figure 8a).
Clearly, Exp3 produces lower rates of biological consumption in the entire basin, showing smaller size of >11 mmol m$^{-3}$ yr$^{-1}$ relative to the reference run, which is located on the southern edge of northern suboxic waters (Figure 8b). Limited modelling studies reported slightly lower rates (<5 mmol m$^{-3}$ yr$^{-1}$) in the mid-waters of tropical Pacific (Llanillo et al., 2018; Shigemitsu et al., 2017). The decrease in biological consumption ranges from ~0.1 to 0.3 mmol m$^{-3}$ yr$^{-1}$ (Figure 8c). For the northern OMZ, biological consumption decreases by ~0.15-0.2 mmol m$^{-3}$ yr$^{-1}$, which is greater than the decreased rate (~0.1
mmol m$^{-3}$ yr$^{-1}$) of physical supply (see Figure 7c).

Biological consumption of DO in the mid-depth water largely results from remineralization of dissolved organic matter (DOM) and decomposition of particulate organic matter (POM). The reference run simulates a range of ~6-12 mmol m$^{-3}$ yr$^{-1}$ and ~0.6-1.5 mmol m$^{-3}$ yr$^{-1}$ for DOM remineralization and POM decomposition, respectively, with the largest values inside





of the suboxic waters (Figure 9a and 9b). Clearly, Exp3 produces lower rates for both DOM remineralization and POM
        decomposition over the entire basin (Figure 9c and 9d), with a decrease of ~0.1-0.2 mmol m$^{-3}$ yr$^{-1}$ and ~0.01-0.02 mmol m$^{-3}$
        yr$^{-1}$, respectively (Figure 9e and 9f). Apparently, the consumption rate contributed by remineralization of DOM is much
        greater than that by decomposition of POM at mid-depth, indicating that remineralization of DOM is the dominant process
        consuming DO at mid-depth. Many other studies also illustrated that the extent of OMZs was not only sensitive to physical
mixing but also to remineralization of DOM (Kriest et al., 2010; Loginova et al., 2019).

### 4.3 Spatial variations of DOM and its remineralization rate

Remineralization rate of DOM in the ocean is determined by the size of DOM pool and temperature (Brewer and Peltzer,
2016; Wang et al., 2008). Given that there is little difference (<10$^{-5}$ °C) in seawater temperature between different model
experiments, the reduced consumption rates due to DOM remineralization would be a result of a smaller size of DOM. Here,
        we evaluate the zonal and meridional distributions distribution of dissolved organic nitrogen (DON) together with
        remineralization rate of DON. As shown in Figure 10a-10d, modelled DON decreases from 6-8 mmol N m$^{-3}$ near the surface
        to 2-4 mmol N m$^{-3}$ over 400-1000 m. Notably, our modelled remineralization rate of DON decreases from ~20 mmol m$^{-3}$ yr$^{-1}$
        in the euphotic zone to ~4-8 mmol m$^{-3}$ yr$^{-1}$ below 400 m, which is similar to the vertical variation of DON. Enhanced
background diffusion leads to a decrease in DON concentration (up to 0.2 mmol N m$^{-3}$) over 300-900 m, but an increase
        below 1000 m in the eastern tropical Pacific (Figure 10e and 10f).

Our previous study showed that the basin-scale model reproduced observed meridional distribution of surface DON (fall
1992) in the eastern tropical Pacific (Wang et al., 2008). A recent field study reported that surface DON concentration was
~5-7 mmol N m$^{-3}$ in the ETSP (Loginova et al., 2019), which is similar to our model results. While there was little
        information on measured rates of DOM remineralization and oxygen consumption at mid-depth of tropical Pacific, there
        were some field studies conducted in other ocean/sections (e.g., subtropical Atlantic and North Pacific), which showed
        comparable or slightly lower rates of oxygen consumption (Sonnerup et al., 2013; Stanley et al., 2012). For instance, cruise
        data within the subtropical North Pacific showed that consumption rate ranged from 8.3 mmol m$^{-3}$ yr$^{-1}$ at ~200 m to <3.1
mmol m$^{-3}$ yr$^{-1}$ below 500 m (Sonnerup et al., 2013).

### 4.4 Impacts of physical supply and biological consumption on asymmetry of OMZs

Previous studies have demonstrated meridional asymmetric features in many physical and biological fields in the tropical
Pacific, e.g., temperature and salinity (Fiedler and Talley, 2006), circulation and ventilation (Kessler, 2006; Kuntz and
Schrag, 2018), nitrogen and carbon cycles (Libby and Wheeler, 1997; Wang et al., 2009), which may be largely associated
with the asymmetries in water mass exchange between the equatorial and off-equator Pacific Ocean (Kug et al., 2003).





Accordingly, one may assume that the hemisphere asymmetry of OMZs could be related to the differences in physical supply and biological consumption between the ETNP and ETSP.

While the dynamics of the tropical Pacific's OMZ is sensitive to changes in ocean circulation (Czeschel et al., 2011; Montes
et al., 2014; Stramma et al., 2010), small scale processes such as background diffusion also have a significant influence on the supply of DO into the core OMZ and the mean state of DO distribution (Duteil and Oschlies, 2011; Getzlaff and Dietze, 2013; Llanillo et al., 2018). Clearly, our model experiment shows that enhanced background diffusion leads to a significant increase in DO concentration below 200 m (Figure 11a). The increase of DO is similar below 600 m in the ETNP and ETSP, but differs largely between the two regions, with much greater values over 200-600 m in the ETSP.


Recent studies have emphasized the role of changes in physical processes for the observed asymmetric OMZs in the tropical oceans. For instance, larger-scale mass transport related to circulation and ventilation in the southern hemisphere is more efficient than in the northern hemisphere (Kuntz and Schrag, 2018), and the transit time from the surface to the OMZ is much longer in the ETNP than in the ETSP (Fu et al., 2018; Sonnerup et al., 2013). Our study shows that enhanced
background diffusion increases the physical supply of DO over most of the water column, with significant increases (>0.4 mmol m$^{-3}$ yr$^{-1}$) below 200 m in the ETSP (Figure 11b). However, the ETNP shows a much smaller increase in physical supply, particular over ~300-700 m (<0.2 mmol m$^{-3}$ yr$^{-1}$) (Figure 11a), indicating that physical supply may be largely responsible for the asymmetry of OMZs. On the other hand, increased DO is greatest at ~700 m in the ETNP (Figure 11a), which cannot be explained by the small increase in physical supply (<0.4 mmol m$^{-3}$ yr$^{-1}$), implying that biological
consumption may also play a role in regulating the dynamics of mid-depth DO.

There is evidence that the size of tropical OMZs is largely influenced by biological processes, such as organic matter export, remineralization and oxygen consumption (Cavan et al., 2017; Keller et al., 2016). Our further analyses demonstrate that biological consumption generally decreases in majority of the water column under enhanced background diffusion, with the
largest decrease (~0.6-0.8 mmol m$^{-3}$ yr$^{-1}$) over ~600-800 m (Figure 11c), which is greater than the increased rate (~0.3-0.5 mmol m$^{-3}$ yr$^{-1}$) of physical supply (Figure 11b). Enhanced background diffusion leads to remarkable decrease of DON over ~300-900 m, with a larger decrease in the ETNP (Figure 11d). Earlier field studies revealed that DON concentration in the euphotic zone is much higher to the north than to the south in the central-eastern tropical Pacific (Libby and Wheeler, 1997; Raimbault et al., 1999). Later studies showed that rates of DOM remineralization and oxygen consumption are also greater at
mid-depth in the ETNP than in the ETSP (Feely et al., 2004; Kalvelage et al., 2015; Tiano et al., 2014). Apparently, the asymmetric OMZs in the tropical Pacific are attributed to not only physical processes, but also biological processes.





## 5. Conclusion

This paper describes an evaluation and validation of a fully coupled basin-scale model (OGCM-DMEC V1.0), focusing on
the sensitivity of the asymmetric OMZs in the tropical Pacific to different parameterizations. Our results show that the
improved model with enhanced background diffusion successfully reproduces the observed asymmetric OMZs in tropical
Pacific, indicating that distribution of DO at mid-depth is more sensitive to physical mixing (e.g., enhanced background
diffusion) rather than biological processes (e.g., reduced DO utilization rate).

Our analyses suggest that DO concentration at mid-depth is significantly increased due to enhanced background diffusion,
resulting from changes in both physical supply and biological consumption. On the one hand, physical supply of DO is
largely increased in the majority of the tropical Pacific, but shows little change at mid-depth in the ETNP. On the other hand,
biological consumption shows a significant decline over 300-1000 m and an increase below 1000 m, as a result of the
redistribution of DOM in the water column.


Our further analyses demonstrate that the asymmetric OMZs in the tropical Pacific are largely associated with asymmetry in
both physical supply and biological consumption. In particular, physical supply plays a dominant role for the asymmetry of
core OMZs (~200-600 m) while biological consumption has impacts on the asymmetric DO below the OMZs, with
implications for the hemisphere asymmetric OMZs. Future studies should consider both physical and biological aspects to
deliver comprehensive assessments on the interactions between DO and DOM, which is critical for a better understanding of
variability and drivers of the tropical asymmetric OMZs.



*Code and data availability.* The exact version of the software code used to produce the results presented in this paper is
archived on Zenodo (http://doi.org/10.5281/zenodo.3890689, Wang et al., 2020). Other code and data are available upon
request from the authors. Request for materials should be addressed to X.J.W. (xwang@bnu.edu.cn).

*Author contributions.* X.J.W. and K.W. designed the study, performed the simulations and prepared the manuscript. R.M.,
D.X.Z. and R.H.Z. contributed to analysis, interpretation of results and writing.

*Competing interests.* The authors declare that they have no conflict of interest.

*Acknowledgements.* This work was supported by the Chinese Academy of Sciences' Strategic Priority Project
(XDA1101010504). The authors wish to acknowledge the use of the Ferret (http://ferret.pmel.noaa.gov/Ferret/).



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





**Figures**

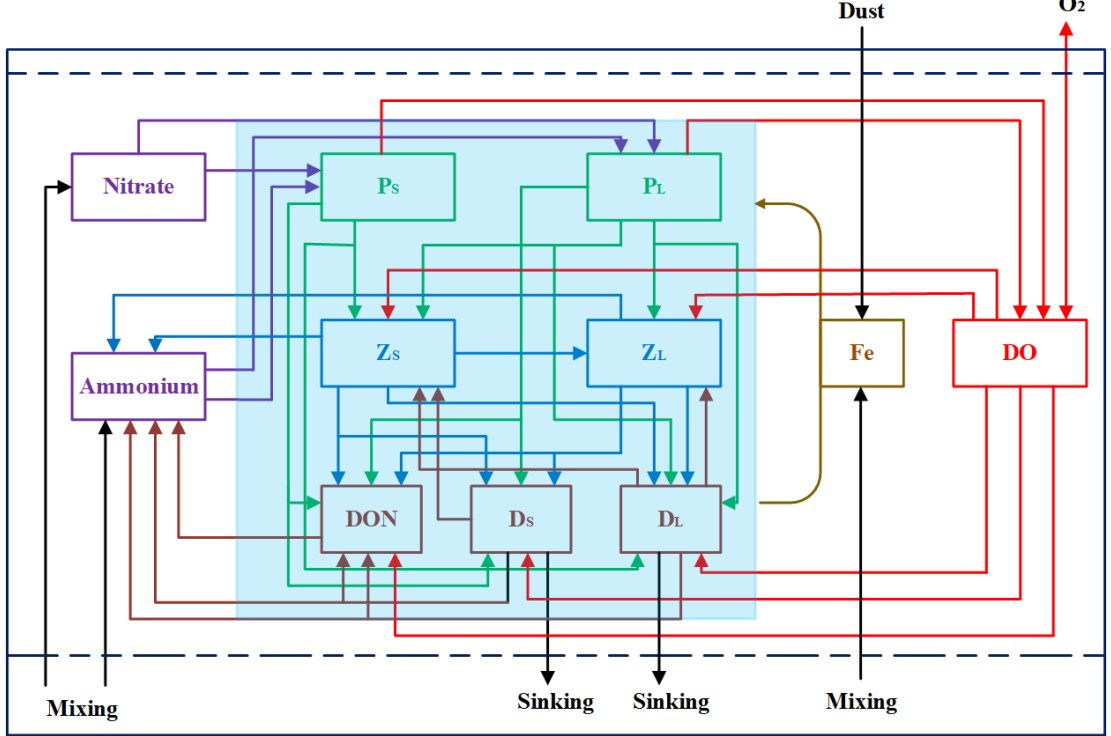

**Figure 1.** Flow diagram of ecosystem model. Red lines and arrows denote oxygen sources and sinks. Purple, green, blue and brown lines and arrows denote fluxes originating from inorganic forms, phytoplankton, zooplankton, and DON and detritus, respectively.

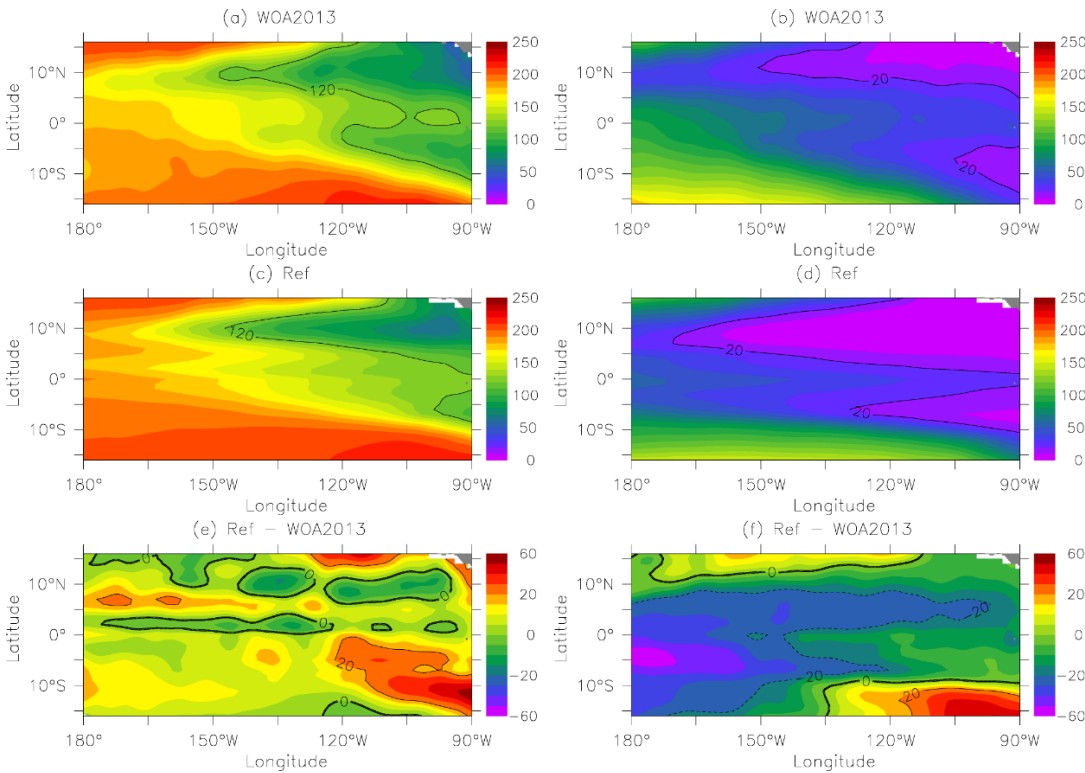


**Figure 2.** Observed and simulated DO concentrations (mmol m$^{-3}$) and the differences over 0-200 m (left panel) and 200-600 m (right panel). **(a)** and **(b)** WOA2013 data, **(c)** and **(d)** reference run for the period of 1981-2000, and **(e)** and **(f)** the difference of DO between model and data. Superimposed black lines denote the borders of low-DO water (<120 mmol m$^{-3}$) in **(a)** and **(c)**, and suboxic water (<20 mmol m$^{-3}$) in **(b)** and **(d)**.




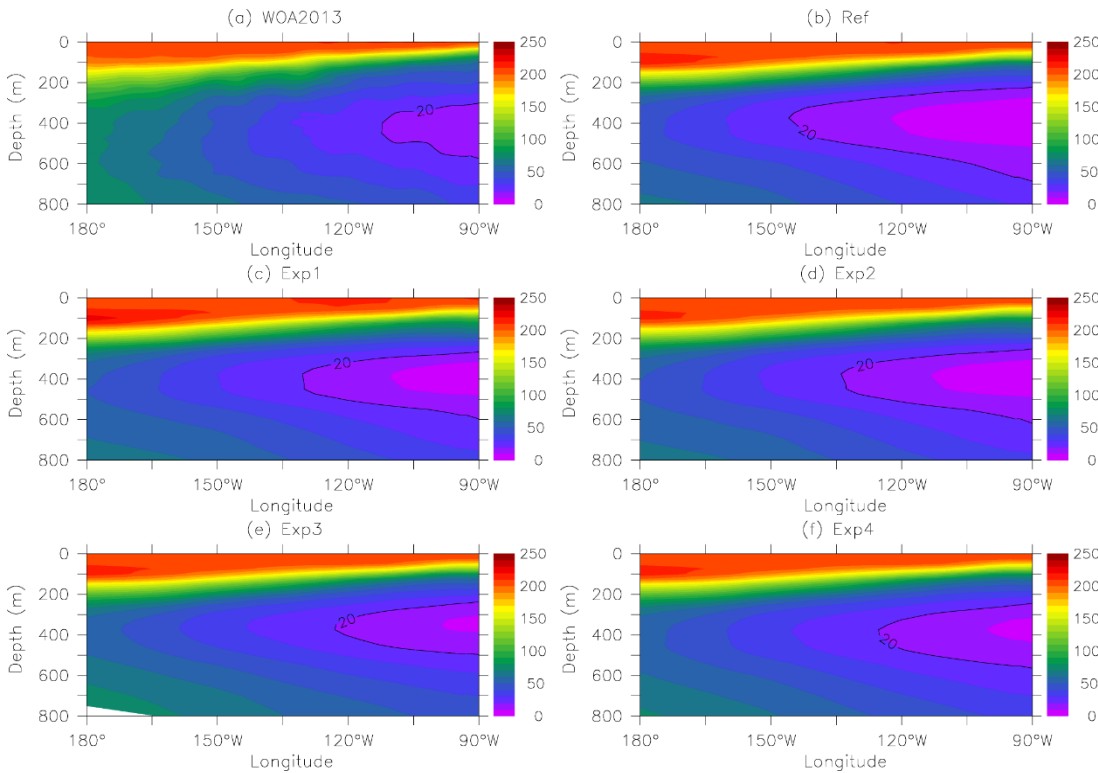

**Figure 3.** Observed and simulated mean DO concentrations (mmol m$^{-3}$) from sensitivity experiments over 10°S-15°N. **(a)** WOA2013 data, **(b)** reference run, **(c)** Exp1, **(d)** Exp2, **(e)** Exp3, and **(f)** Exp4 for the period of 1981-2000. Superimposed black lines denote the borders of suboxic water (<20 mmol m$^{-3}$) in **(a-f)**.




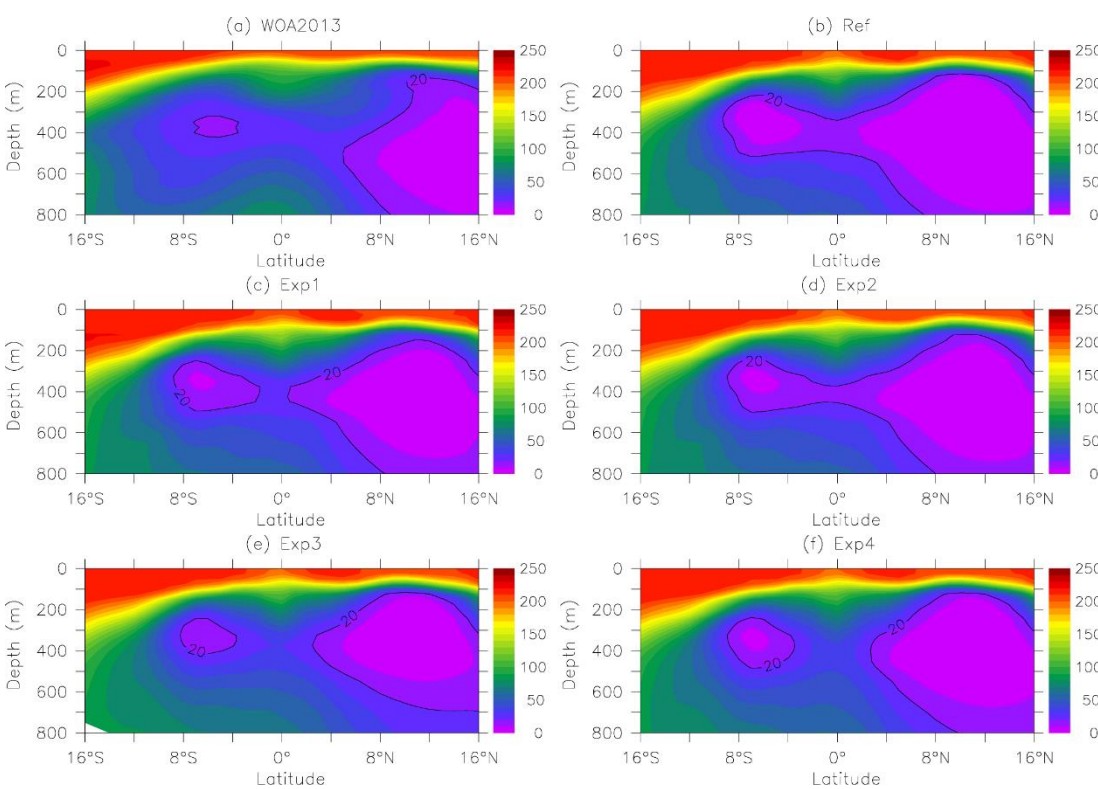

**Figure 4.** Observed and simulated mean DO concentrations (mmol m$^{-3}$) from sensitivity experiments over 130°W-90°W. **(a)** WOA2013 data, **(b)** reference run, **(c)** Exp1, **(d)** Exp2, **(e)** Exp3, and **(f)** Exp4 for the period of 1981-2000. Superimposed black lines denote the borders of suboxic water (<20 mmol m$^{-3}$) in **(a-f)**.





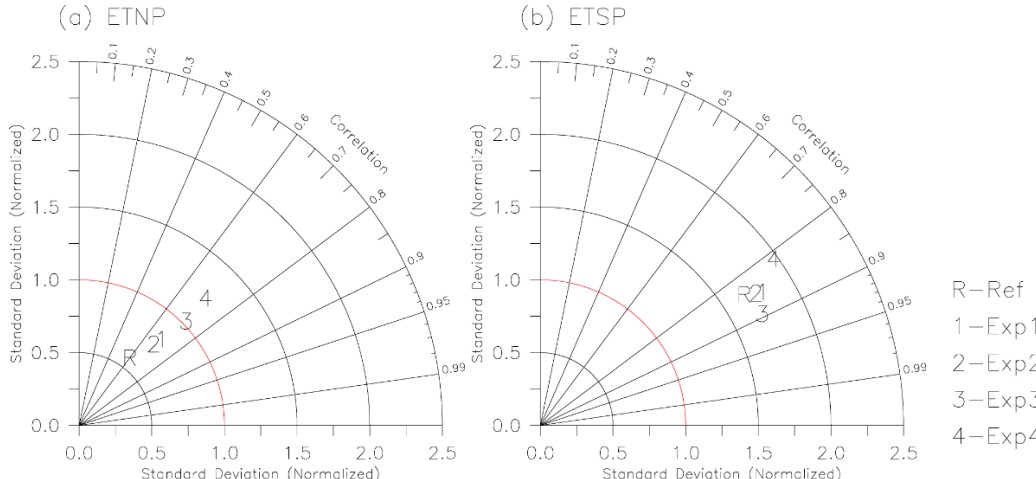

**Figure 5.** Taylor diagram for reference run and sensitive experiments comparing with WOA2013 data. Diagrams are shown for the two regions **(a)** ETNP and **(b)** ETSP.

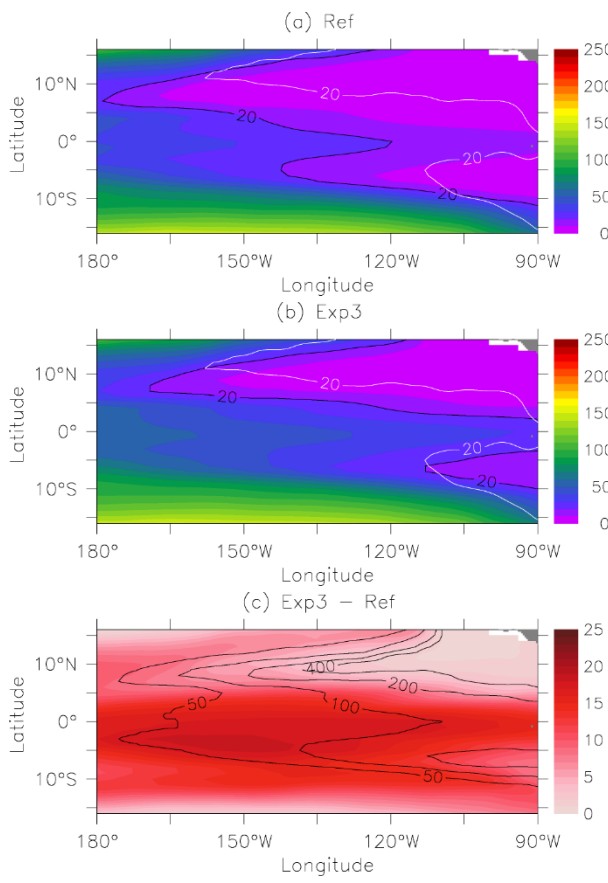


**Figure 6.** Modelled mean DO concentrations (mmol m$^{-3}$) over 300-500 m. **(a)** reference run, **(b)** Exp3, and **(c)** the difference between the means. Superimposed white and black lines in **(a)** and **(b)** denote the borders of suboxic water (<20 mmol m$^{-3}$) from WOA2013 and model simulations, respectively. Superimposed solid black lines in **(c)** denote the percentage of DO change.




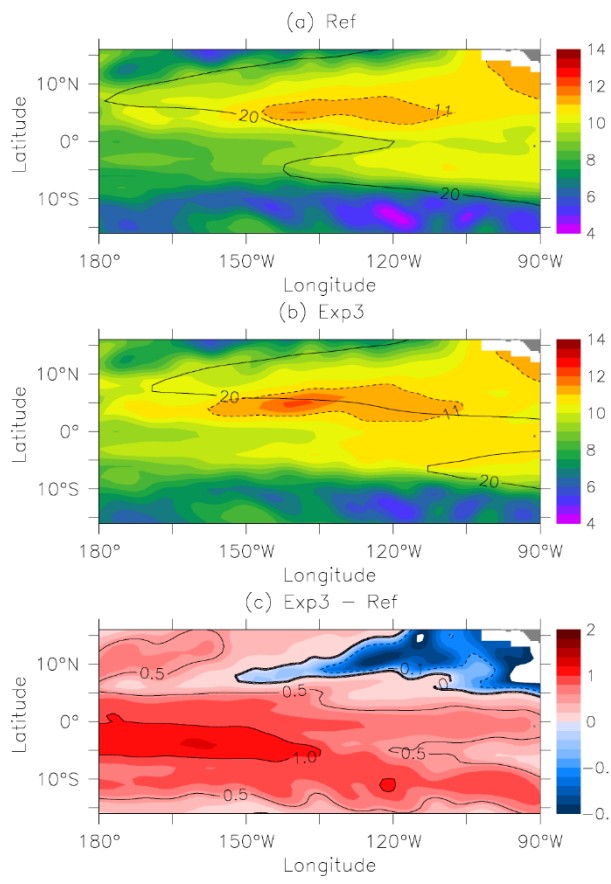

**Figure 7.** Physical supply (mmol m$^{-3}$ yr$^{-1}$) over 300-500 m. (a) reference run, (b) Exp3, and c the difference between the means. Superimposed solid black lines in (a) and (b) denote the borders of suboxic water (<20 mmol m$^{-3}$) from simulations.

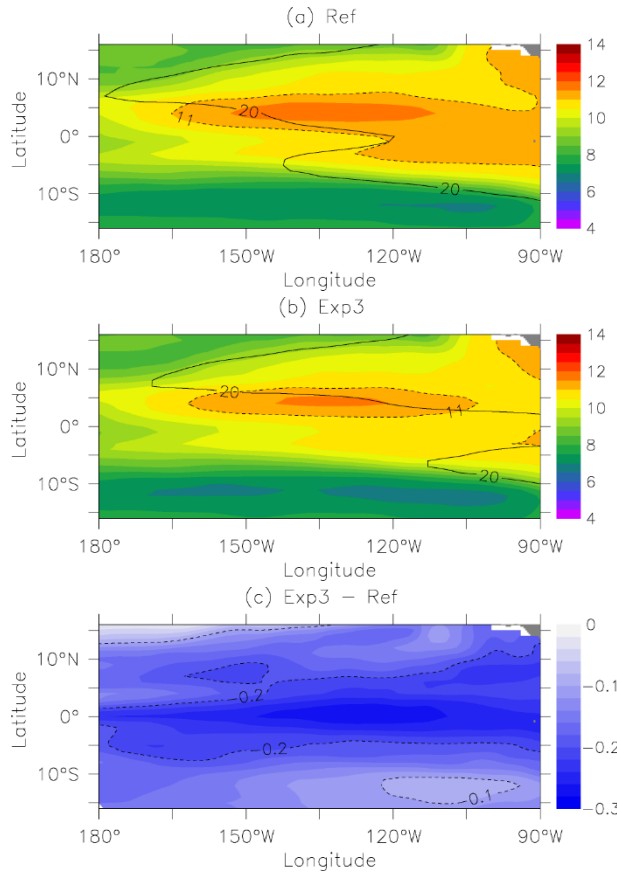


**Figure 8.** Biological consumption (mmol m$^{-3}$ yr$^{-1}$) over 300-500 m. **(a)** reference run, **(b)** Exp3, and **(c)** the difference between the means. Superimposed solid black lines in **(a)** and **(b)** denote the borders of suboxic water (<20 mmol m$^{-3}$).



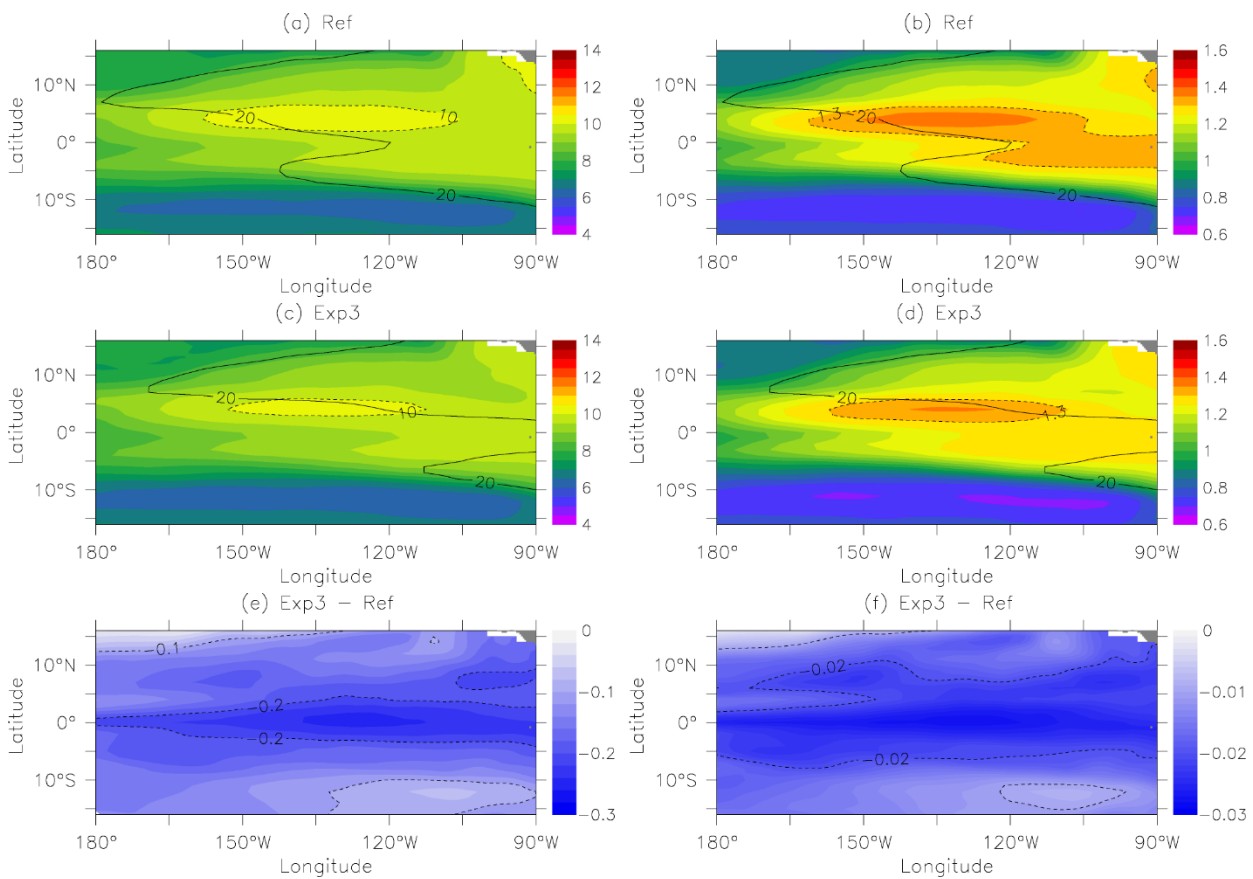

**Figure 9.** DOM remineralization (left panel, mmol $m^{-3}$ $yr^{-1}$) and POM decomposition (right panel, mmol $m^{-3}$ $yr^{-1}$). **(a)** and **(b)** reference run, and **(c)** and **(d)** Exp3, and **(e)** and **(f)** the difference between the means. Superimposed solid black lines in **(a-d)** denote the borders of suboxic water (<20 mmol $m^{-3}$).





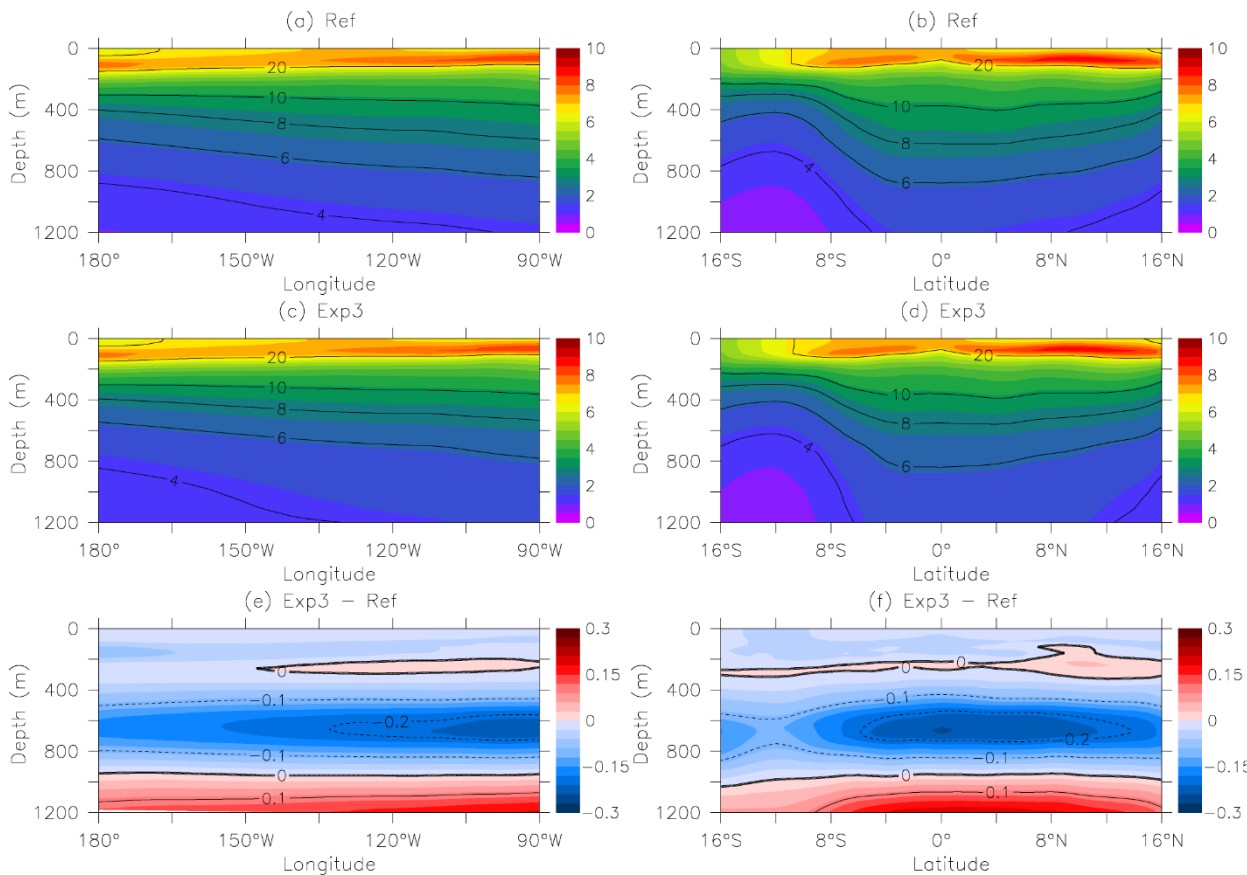

**Figure 10.** Mean DON concentrations (mmol N m$^{-3}$) and DON remineralization (mmol m$^{-3}$ yr$^{-1}$) over 10°S-15°N (left panel) and 130°W-90°W (right panel). **(a)** reference run, **(b)** Exp3, and **(c)** the difference between the means. Superimposed black lines in **(a-d)** denote consumption rates (mmol m$^{-3}$ yr$^{-1}$) by remineralization of DON.





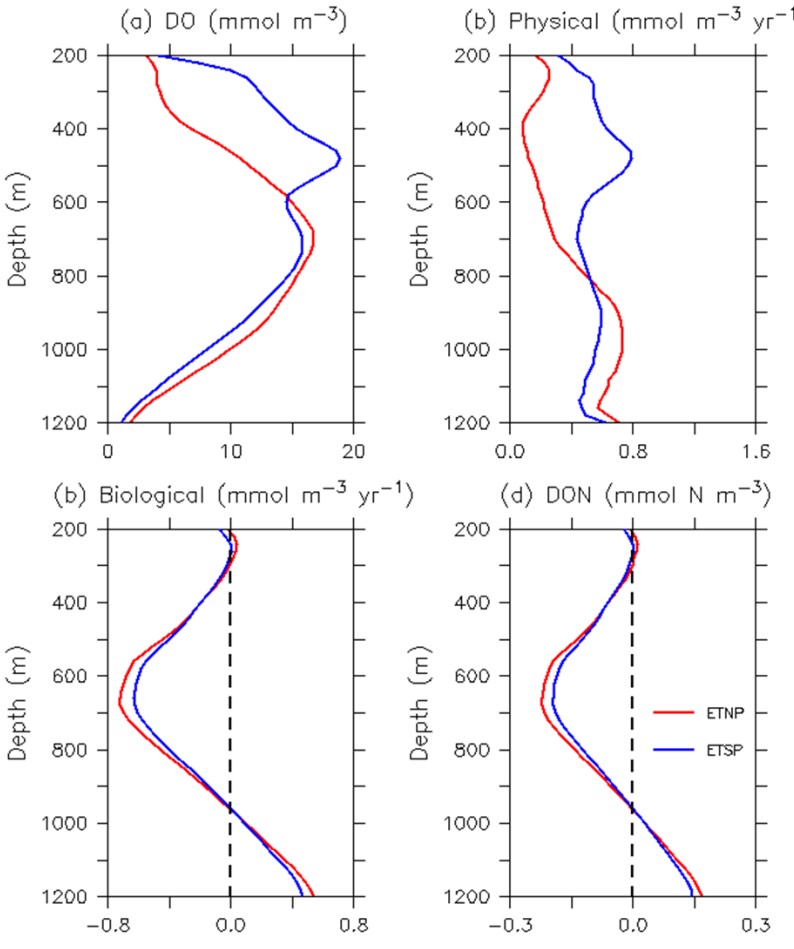

**Figure 11.** Comparisons between Exp3 and reference run (Exp3 minus reference run). **(a)** DO, **(b)** physical supply, **(c)** biological consumption, and **(d)** DON. Sections: ETNP (red lines) (150°W-90°W, 0°N-15°N) and ETSP (blue lines) (150°W-90°W, 10°S-0°S)





## Tables

**Table 1.** Model experiments with different parameters in biological and physical processes.

| Parameter | Symbol | Unit | Ref | Exp1 | Exp2 | Exp3 | Exp4 |
|-----------|--------|------|-----|------|------|------|------|
| Remineralization | $C_{DON}$ | $d^{-1}$ | 0.001 | 0.0005 | 0.001 | 0.001 | 0.001 |
| O:C ratio | $R_{OC}$ | - | 1.3 | 1.3 | 1.0 | 1.3 | 1.3 |
| Partial mixing | $P_m (R_i>0.7)$ | $m^{-2} s^{-1}$ | 0 | 0 | 0 | $10^{-4}$ | $\frac{3.5 \times 10^{-3}}{(0.7 + Ri)} + 3 \times 10^{-5}$ |

Ri is Richardson number in the water column.

**Table 2.** Comparisons of DO over 300-500 m between WOA2013 data and model experiments for the period of 1981-2000 in the Eastern Tropical North Pacific (ETNP) and Eastern Tropical South Pacific (ETSP).

| | Ref | Exp1 | Exp2 | Exp3 | Exp4 |
|---|---|---|---|---|---|
| **ETNP (150°W-90°W, 0°N-15°N)** | | | | | |
| Bias (mmol m$^{-3}$) | -16.13 | -11.51 | -12.71 | -8.59 | -9.55 |
| RMSE (mmol m$^{-3}$) | 18.43 | 14.39 | 15.34 | 12.36 | 13.73 |
| Correlation coefficient | 0.599* | 0.691** | 0.675** | 0.715** | 0.705** |
| NSD | 0.581 | 0.820 | 0.761 | 1.027 | 1.234 |
| | | | | | |
| **ETSP (150°W-90°W, 10°S-0°S)** | | | | | |
| Bias (mmol m$^{-3}$) | -14.70 | -7.09 | -8.96 | 1.05 | -2.71 |
| RMSE (mmol m$^{-3}$) | 15.51 | 9.87 | 11.04 | 8.43 | 9.69 |
| Correlation coefficient | 0.842** | 0.857*** | 0.852*** | 0.893*** | 0.814** |
| NSD | 1.666 | 1.784 | 1.740 | 1.709 | 1.971 |

Significances of correlation coefficient at 0.05, 0.01, and 0.001 levels are marked with one, two and three asterisks, respectively.