# Peer review of "Evaluation of asymmetric Oxygen Minimum Zones in the tropical Pacific: a basin-scale OGCM-DMEC V1.0"

_Geoscientific Model Development, 2020_

## Referee Comment (RC1) · Anonymous Referee #1 · 7 Sep 2020

The manuscript investigates, using a model, potential processes that can explain the asymmetry of the tropical Pacific Ocean Oxygen Minimum Zones (OMZs). The topic is important as the modelling of the Tropical Pacific OMZ, and, in particular its asymmetry, is a challenge that have to face modelers. However, in its present form, I have serious reservations about the scientific significance of this manuscript. Essentially, the work presents the results of 4 experiments performed with a coupled physical biogeochemical model. The paper is not innovative in terms of modelling, the authors refer to another work for the description and validation of the model while the analysis of model results is quite basic and could have been done with much more details.

Starting from an initial parameterization of the model (reference simulation) that gives results that are broadly validated with the WOA 2013 climatology, the authors decide

to perform 4 experiments in which they change the degradation parameter (2 experiments) and vertical (diffusive) mixing (2 experiments) in order to better simulate the volume of low oxygen zone in the region of the Tropical Pacific. Then, the authors compare the 5 simulations and conclude that an increase in the vertical mixing helps with representing the asymmetry in the Tropical OMZ.

First, there are few rationales for justifying the choice and new formulations of the investigated processes (i.e; mixing, degradation). The physics and other biogeochemical variables are not shown and hence for the reader this is not straightforward to understand what motivates the authors to believe that the mixing and degradation are the process that need to be improved. They do not show evidences that the model overestimates degradation or underestimates mixing when looking at modeled variables. Then, the authors do not investigate what are the consequences for the simulated physics and biogeochemistry of such changes. Rather, the different experiments are compared with climatology but only for oxygen and over 300-500m depths. The authors do not mention how an increase in diffusion and transport of oxygen will impact oxygen in the layer above 200 m and below 500m neither the consequences of this increased diffusion for the other variables (physical and biogeochemical) in terms of agreement with observations.

As important, in terms of biogeochemical modelling, the authors decide not to describe the model and to refer to Wang et al (2008) for details. However, looking at Wang et al (2008), I was not able to find oxygen as a state variable which means that the modeling of oxygen is not described neither its validation which is an important prerequisite before starting sensitivity studies. I am surprised to see that important process like nitrification or oxygen production associated with nitrate reduction are not taken into account. I would have hoped to see a detailed description of the modeled oxygen cycling and model formulations with a thorough validation of model performances using oxygen data (in addition to a very board comparison with climatology). This comparison would have allowed the reader to clearly understand model limitations and reasons

for changing model formation. Besides, the resolution of the model as well as that of the forcing (i.e. 6-day averaged mean wind stress) is quite rough and this may also explain some of the model deficiencies but this is not discussed at all.

Finally, the plausibility of the sensitivity studies is not discussed. I was just wondering what are the rationales for using a background diffusion that is 100 times higher than molecular diffusion and using a modified O:C ratio,

Details Line 14 For clarity DO needs to be defined, ETNP

Line 28: "The carbon cycle has garnered much attentions and made significant process", This sentence should be rewritten e.g. The carbon cycle has garnered much attentions and its understanding made significant progresses

Line 29: physical/chemical processes (e.g., the fluxes between the atmosphere, land and ocean). This is vague please specify

Line 29: in most ocean basin, DO concentration is not below 20 mmol /m3 except in OMZs of the Pacific and Indian Ocean

Line 56: Please specify: "missing biogeochemical feedbacks in the models".

Line 77: "Chen mixing scheme (Chen et al., 1994), which varies from 10 m to 50 m on the equator." I assume that it is not the mixing scheme that varies between 10m to 50 m but rather the vertical resolution. Correct?

Line 78: what is the vertical resolution in the OMZs ?

Line 8 is it evaluated using the simulated SST or at 20°C ? I do not understand why we have "at 20°C)

Line 162-162: "some models overestimated the extent of suboxic water, which might be due to over-estimated productivity in the euphotic zone" This conclusion does not seem in agreement with results of Exp1 and Exp2 that show that a decrease in respiration does not allow the representation of asymmetric OMZ

Line 225: I find that the use of smaller size is confusing. I guess that the authors mean smaller amount. (and not particles size since the DOM is dissoved).

Line 245: the authors mention that the asymmetric features in many physical and biological fields in the Tropical Pacific are largely associated with asymmetries in water mass exchange between the equatorial and off-equator Pacific Ocean. However, here they use an enhanced vertical diffusion to create this asymmetry and this is not clear how this parameterization can mimic asymmetry water masse exchanges with the regional outside the Pacific.

Section 3 (very broadly) describes the results of the experiments but is placed outside the results section.

Figure 5: I would say sensitivity experiments rather than sensitive experiments.

Table 3: please correct Pm is a diffusion coefficient and has to be in m2/sec and not /m2/sec.

[Figure]

---

## Referee Comment (RC2) · Anonymous Referee #2 · 29 Sep 2020

**Subject**

Wang and co-authors investigate the impacts of physical supply and biological consumption of dissolved oxygen (DO) on the dynamics and asymmetry of the OMZs in the tropical Pacific.They perform 4 numerical experiments to evaluate the sensibility of the mid-depth oxygen concentration to these aspects in their model, OGCM-DMEC v1.0. The physical supply is evaluated through the background diffusion parameterisation (that the authors test by changing a partial mixing parameter) and the effects of biological consumption on oxygen are tested by changing the C:O utilization ratio. The final aim is to advance their model capacity to simulate the oceanic oxygen cycle, and to explore the mechanisms driving the asymmetric OMZs in the tropical Pacific (introduction, l.67-68).

Relevance of the subject

To understand the physical and biological processes responsible for the asymmetry of tropical Pacific OMZs is a topic of great interest for climate modelers, which has currently not been solved.

General comments

However, in its present form, the conclusions of this study bring no new clues of understanding, and do not explore any mechanisms. The authors conclude that both physical supply and biological consumption impact the OMZs extend and vertical structure, which, according to them, has been the subject of numerous previous papers (see l.188-190 or l.219-220).

While it is a promising approach to explore the DO budget term by term, I recommend to enlarge the analyses to other variables (by characterizing the tropical ocean dynamics with vertical sections of horizontal currents for example, and by giving insights of plankton and nutrients mean-state and variability) in order to explore the mechanisms at play when increasing the background diffusion or decreasing the biological consumption.

Besides I have some reservations about the use of a basin-scale model of the Pacific limited at 20°S and 20°N to study the Pacific OMZs. It seems not very appropriated to model OMZ borders, as these latter are found far north of 20°N. If the aim of the study is to investigate the importance of DO physical supply, one may not ignore the ventilation processes at play in the OMZ borders (Bettencourt et al., 2015). And even in a case of tropical study (as reflected by the analyses restricted to 15°S-15°N), one may not ignore the critical representation of the equatorial undercurrent (EUC) to model the tropical OMZ structure (Busecke et al., 2019). As both processes are highly resolution-dependent (see for example Fig. 16 in Berthet et al., 2019), I am surprised to find no discussion and no bibliography on the questions of the appropriate model resolution needed to get a realistic OMZ structure.

[Figure]

Results

Model description: The parameterization of the oxygen cycle needs to be described with more details. It would help the reader to analyse the results.

Validation: In its present form, the model validation may be completed by showing physical currents and temperature/salinity mean state and variability (at the surface and with a vertical section along latitudes), OMZ inter-annual variability, ventilation at the OMZ boundaries (as mesoscale activity has been shown to shape the OMZ)...

As stated l.70, the OGCM-DMEC V1.0 has shown a good model-data agreement in the carbon cycle for the tropical Pacific Ocean (Wang et al., 2015). This is a good point if the model was validated on carbon cycle, but the paper needs a true validation on oxygen.

Specific comments

l.14: 'DO' is used in the abstract, but not defined

l.53: I would recommend to add the following study to justify that circulation play a dominant role in regulating the dynamics of tropical OMZs: Busecke, J. J. M., Resplandy, L., & Dunne, J. P. P. (2019). The Equatorial Undercurrent and the oxygen minimum zone in the Pacific. Geophysical Research Letters, 46. https://doi.org/10.1029/2019GL082692

l.54-57: And what about the horizontal resolution of the model ? Using an ESM with a high-resolution ocean (1/10°), Busecke et al. (2019) show that a realistic representation of the Equatorial Undercurrent (EUC) dynamics is crucial to represent the upper OMZ structure and its temporal variability. They demonstrate that coarser ESMs commonly misrepresent the EUC, leading to an unrealistic "tilt" of the OMZ (e.g., shallowing toward the east) and an exaggerated sensitivity to EUC changes overwhelming other important processes like diffusion and biology.

This last aspect would be interesting for your discussion.

l.61: "A fully coupled basin-scale physical-biogeochemical model (OGCM-DMEC V1.0) was developed for the tropical Pacific (Wang et al., 2008; Wang et al., 2015; Wang et al., 2009)." –> are you using a regional configuration centered on the Pacific ocean ? Or is it a global model ?

l.78: "The model domain is between 30°S and 30°N" –> thus it is not "global". This has to be clarified, as OGCM generally means ocean GLOBAL circulation model.

Moreover if your domain extends between 30°S-30°N: why did you crop your horizontal maps at 15°N while Fig. 2b clearly not catch OMZ northern border between 200-600m (which seems far north) ? I would suggest to enlarge the northern border up to 20°N (at least, as your sponge layers are in the 20°-30° bands).

"and zonal resolution is 1°." –> have you checked how your EUC behaves ?

l.86-87: precipitation (gpcp) and wind stress (NCEP) forcings are not consistent ?

l.90: "an interannual simulation for the period of 1978-2000, and analyze model output for the period of 1981-2000." –> could you give some insights about the interannual behaviour of your OMZs ?

l.96: DON is not defined

l. 107-109: please clarify your computation of the vertical mixing term: "the vertical mixing term that is calculated by three subroutines (Chen et al., 1994)." –> I guess that to be splitted in 3 subroutines is not the main characteristic of the hybrid scheme of Chen et al. (1994). It would be interesting to mention why you add this mixing scheme in your model from a physical point of view. Following the abstract of Chen et al. (1994), this hybrid vertical mixing scheme "helps to produce more realistic velocity profiles in the eastern and central equatorial Pacific. This is mainly due to the improved parameterization of interior mixing related to the large shears of the Equatorial Under-current", which seems to me an important aspect when modelling the OMZ. Or it would be important to tell the reader (still from their abstract) that this scheme "is capable of

simulating the three major mechanisms of vertical turbulent mixing in the upper ocean, that is, wind stirring, shear instability, and convective overturning."

l. 137: as you aim to determine the respective roles of physics and biogeochemistry in the oxygen biases, it would be helpful to have some basic validations on horizontal / vertical circulations (for example, a vertical section of zonal jets along the latitudes) and nutrients affecting the oxygen budget in your model (phyto- and zoo-plankton, detritus, DON, ammonium, nitrate).

l.142-145: Regarding your sensitivity experiments, it would be helpful to clarify how the initial DON remineralization constant and O:C utilization ratio were determined.

Moreover, are you increasing the oxygen supply through mixing only in the OMZ region or in the whole Pacific basin ? Could you justify your choices ? Could you elaborate on your "variable Pm" ? How does it vary ?

l.179-180: "We first compare the distribution of DO over 300-500 m between reference run and Exp3. The reference run produces much large volume of suboxic waters (<20 mmol m-3) in both the ETNP and ETSP where the two OMZs are merged (Figure 6a)."

The reader would appreciate if the oxygen average for your "ref" experiment in Fig. 6 may be comparable with observations: Fig. 2 (right column) shows the 200-600 m mean, and Fig. 6 the 300-500 m mean. These 2 averaged layers (200-600 vs 300-500 m) are quite different in terms of volume of equatorial suboxic waters, so, please, could you add a 3rd column in Fig. 2 with the 300-500 m mean in WOA ?

l. 181: "Exp3 performs well in reproducing the sizes and locations of two asymmetric OMZs" –> the use of quantitative metrics (OMZ volume, maximal horizontal extent) would reinforce this conclusion.

l.195: regarding the small decrease you detect in the ETNP-OMZ in exp3 (Fig. 7c): what do you obtain with exp4 ? Is this decrease linked with coastal processes ? If yes, how ?

Figures

Figure 1: legend of Ps, PL, Zs, ZL, Ds, DL is missing.

Figure 2: it seems weird to me to study the Pacific OMZ but to not catch its spatial extend entirely: why don't you extend your simulated regions at least to 25°S and 25°N (shifting your sponge layers between 30° and 35° for example), and to the coasts of America (~70°W to get both northern and southern parts of the Pacific OMZ) ?

Figures 3 (and 10): as the paper focus on the asymmetry between the northern and southern part of the Pacific OMZ, and as its aim is to show how they differ, the meridional means between 10°S-15°N seem not appropriate. I would recommend to split the analyse in two, one for each OMZ (south and north). As it is, Fig. 3 does not allow to properly evaluate how the model reproduces the vertical structure of the OMZ against observations.

Same comment for Fig. 10 (left column), and this analyse does not allow to investigate any mechanisms.

Bibliography

Berthet, S., Séférian, R., Bricaud, C., Chevallier, M., Voldoire, A., & Ethé, C. (2019). Evaluation of an online grid‐coarsening algorithm in a global eddy‐admitting ocean biogeochemical model. Journal of Advances in Modeling Earth Systems, 11, 1759–1783. https://doi.org/10.1029/2019MS001644

Bettencourt, J. H., López, C., Hernández‐García, E., Montes, I., Sudre, J., Dewitte, B., et al. (2015). Boundaries of the Peruvian oxygen minimum zone shaped by coherent mesoscale dynamics. Nature Geoscience, 8, 937–940. https://doi.org/10.1038/ngeo2570

Busecke, J. J. M., Resplandy, L., & Dunne, J. P. P. (2019). The Equatorial Undercurrent and the oxygen minimum zone in the Pacific. Geophysical Research Letters, 46. https://doi.org/10.1029/2019GL082692

---

## Author Comment (AC1) · 30 Nov 2020

**Response to Anonymous Referee #1**

The manuscript investigates, using a model, potential processes that can explain the asymmetry of the tropical Pacific Ocean Oxygen Minimum Zones (OMZs). The topic is important as the modelling of the Tropical Pacific OMZ, and, in particular its asymmetry, is a challenge that have to face modelers. However, in its present form, I have serious reservations about the scientific significance of this manuscript. Essentially, the work presents the results of 4 experiments performed with a coupled physical biogeochemical model. The paper is not innovative in terms of modelling, the authors refer to another work for the description and validation of the model while the analysis of model results is quite basic and could have been done with much more details.

**Response:** Thank you for the constructive comments. We have made major revisions to improve our manuscript. For example, we have added more details in model description, and more information regarding model experiments. We have also revised our approach in terms of model validation, and the analyses of model results with much more details.

Starting from an initial parameterization of the model (reference simulation) that gives results that are broadly validated with the WOA2013 climatology, the authors decide to perform 4 experiments in which they change the degradation parameter (2 experiments) and vertical (diffusive) mixing (2 experiments) in order to better simulate the volume of low oxygen zone in the region of the Tropical Pacific. Then, the authors compare the 5 simulations and conclude that an increase in the vertical mixing helps with representing the asymmetry in the Tropical OMZ.

First, there are few rationales for justifying the choice and new formulations of the investigated processes (i.e; mixing, degradation). The physics and other biogeochemical variables are not shown and hence for the reader this is not straightforward to understand what motivates the authors to believe that the mixing and degradation are the process that need to be improved. They do not show evidences that the model overestimates degradation or underestimates mixing when looking at modeled variables.

**Response:** Thank you for the constructive comments. We have made the following changes:

(1)  We have added more details about the model in the section 2.2 Ocean biogeochemical model: "The equations for biogeochemical processes and model parameters are described in Appendix A and B. There have been changes in some parameters comparing with those in Wang et al. (2008), which were based on our model calibration and validation for chlorophyll (Wang et al., 2009a, Wang et al., 2013), nitrogen cycle (Wang et al., 2009b) and carbon cycle (Wang et al., 2015)". We have also provided more information in the section 2.3 Computation of oxygen sources and sinks, e.g., "Below the euphotic zone, the concentration of DO is influenced by physical supply and biological consumption …".

(2)  We have revised/rewritten the section 3.2 Sensitivity experiments. In particular, to clarify the rationales for the model experiments, the first paragraph has been rewritten as "Given that the mid-depth DO concentration is influenced by physical supple and biological consumption, and remineralization of DON is the dominant process for oxygen consumption, the underestimated DO at mid-depth would be a result of overestimation of consumption associated with DON

remineralization and/or underestimation of supply. Indeed, the reference run over-estimates
biological consumption over 100-400 m (Figure 3). Thus, we apply a reduced (by 50%) DON
remineralization constant, which leads to a remarkable improvement in simulated DON and
consumption. The reference run applies a zero value for background diffusion (equation 4).
However, a previous modeling study has demonstrated that background diffusion is an
important process for DO supply at mid-depth (Duteil and Oschlies, 2011). Accordingly, we
conduct a few more simulations to investigate how reduced remineralization rate and adding
background diffusion affect simulated DO distribution, which include changing the partial
mixing parameter Pm from 0 to 0.1, 0.3, 0.5 or 1.0 cm$^2$ s$^{-1}$ (Table 1)".

(3) We have also reorganized the sensitivity experiments, which include some new simulations,
in response to some other comments (see more information/responses below).
Then, the authors do not investigate what are the consequences for the simulated physics and
biogeochemistry of such changes. Rather, the different experiments are compared with
climatology but only for oxygen and over 300-500 m depths. The authors do not mention how an
increase in diffusion and transport of oxygen will impact oxygen in the layer above 200 m and
below 500 m neither the consequences of this increased diffusion for the other variables (physical
and biogeochemical) in terms of agreement with observations.

**Response:** Thank you the constructive comments. We have showed the comparisons of DO over
200-400, 400-700 and 700-1000 m, and also added more model-data comparisons using cruises'
DO data. We have added a new figure to show the impacts of reduced remineralization and
increased diffusion on the vertical distributions of DON and oxygen consumption in terms of
agreement with observations (see figure below). In addition, we have revised the discussion
section, with new figures to show the changes in DO, biological consumption and physical supply
over 200-400, 400-700 and 700-1000 m.

[Figure]

**Figure 3.** Comparisons of DON concentration (a) and consumption rate (b) between observation and
model experiments. Observed DON data are from Hawaii Ocean Time-series program (HOT, 22°45'N,
158°00'W) (https://hahana.soest.hawaii.edu/hot/hot_jgofs.html). Observed consumption data are obtained
from Karastensen et al., (2008) for the entire Pacific.

As important, in terms of biogeochemical modelling, the authors decide not to describe the model
and to refer to Wang et al (2008) for details. However, looking at Wang et al (2008), I was not
able to find oxygen as a state variable which means that the modeling of oxygen is not described
neither its validation which is an important prerequisite before starting sensitivity studies. I am
surprised to see that important process like nitrification or oxygen production associated with
nitrate reduction are not taken into account. I would have hoped to see a detailed description of
the modeled oxygen cycling and model formulations with a thorough validation of model
performances using oxygen data (in addition to a very board comparison with climatology). This
comparison would have allowed the reader to clearly understand model limitations and reasons
for changing model formation. Besides, the resolution of the model as well as that of the forcing
(i.e. 6-day averaged mean wind stress) is quite rough and this may also explain some of the
model deficiencies but this is not discussed at all.

**Response:** Thank you for the constructive comments. This basin-scale model was developed to
study the upper ocean dynamics for the tropical Pacific, and has been used to understand the
spatial and temporal variability of physical and biogeochemical processes. Our previous studies
have shown that this model can reproduce mesoscale and sub-mesoscale structures such as the
tropical instability wave (TIW) (Zhang, 2016; Zhang and Busalacchi, 2008), and the carbon
model (Wang et al., 2015) forced by 6-day mean winds did a good job in simulating the carbon
fields in the Tropical Pacific. Thus, we believe that the potential bias caused by the resolution of
our model and 6-day winds would be small.

The model does incorporate nitrification (see Wang et al., 2009b). There have been
changes/improvements (relative to Wang et al. 2008) in some parameters, which were based on
our further model calibration and validation, mainly for chlorophyll (Wang et al., 2009a; Wang et
al., 2013), nitrogen cycle (Wang et al., 2009b) and carbon cycle (Wang et al., 2015). Oxygen is a
state variable in the basin-scale biogeochemical model, but this is the first time showing mode
calibration and validation for oxygen cycle. Most parameters used to compute the sources/sinks
of oxygen are the same as those for nitrogen and carbon cycles. We agree with that more details
about the model and more model validation should be presented in this paper. We have added
model equations and parameters, and carried out more model-data comparisons.
Finally, the plausibility of the sensitivity studies is not discussed. I was just wondering what are
the rationales for using a background diffusion that is 100 times higher than molecular diffusion
and using a modified O:C ratio,

**Response:** Thank you for the constructive comments. We have made major revisions, with more
information regarding the sensitivity studies (see responses above). We realize that there was
some "weakness" in our previous model experiments, e.g., no combination of reduced
remineralization and enhanced background diffusion. Thus, we have conducted some new
experiments, including testing different values for background diffusion.

There is a large range (~0.01-0.5 $cm^2/s$) in the parameter for background diffusion ($K_b$) used in
modeling studies. It appears that smaller values are used in ocean models that apply the KPP
scheme (Large et al., 1994) but higher values used in models with other mixing scheme. For
example, Zhu and Zhang (2018) used 0.01 $cm^2/s$ in an ocean model that has the KPP scheme, but

Wang and Matear (2001) used 0.1 cm²/s in a model with the Chen mixing scheme (Chen et al.,
1994). Wang (2002) conducted a comparison of the Chen scheme ($K_b$=0.1 cm²/s) and KPP
scheme ($K_b$=0), which showed large similarity in SST, SSS and MLD (see Figure below).

[Figure]

**Figure 2.3** The daily sea surface temperature (SST), sea surface salinity (SSS), and mixed layer depth (MLD) from the simulations by the Chen scheme (dark line) and the KPP scheme (light line) and observation for the period of September 1996 to December 1997 in the Subantarctic Zone of the Southern Ocean.

Wang, Xiujun (2002). Modeling upper ocean dynamics in the Southern Ocean: Interaction of
physics and biogeochemistry. Ph.D. Thesis, page 31, University of Tasmania.

Details Line 14 For clarity DO needs to be defined, ETNP

**Response:** We have defined DO and ETNP.

Line 28: "The carbon cycle has garnered much attentions and made significant process", This
sentence should be rewritten e.g. The carbon cycle has garnered much attentions and its
understanding made significant progresses

**Response:** We have reworded as "the carbon cycle has garnered much attentions, which made
significant progresses".

Line 29: physical/chemical processes (e.g., the fluxes between the atmosphere, land and ocean).
This is vague please specify

**Response:** We have reworded as "physical/chemical processes (e.g., carbon fluxes between the
atmosphere, land and ocean)".

Line 39: in most ocean basin, DO concentration is not below 20 mmol /m3 except in OMZs of
the Pacific and Indian Ocean

**Response:** Thank you for the constructive comments. We have deleted that part of the sentence.

Line 56: Please specify: "missing biogeochemical feedbacks in the models".

**Response:** Thank you for the suggestion. We have rewritten this sentence.

Line 77: "Chen mixing scheme (Chen et al., 1994), which varies from 10 m to 50 m on the
equator." I assume that it is not the mixing scheme that varies between 10m to 50 m but rather the
vertical resolution. Correct?

**Response:** We have corrected as "The mixed layer (the upper-most layer) depth is determined...,
which varies from 10 m to 50 m".

Line 78: what is the vertical resolution in the OMZs ?

**Response:** The vertical resolution is ~30-50 m in the core OMZ.

Eq. 8: is it evaluated using the simulated SST or at 20 C ? I do not understand why we have "at
20C)

**Response:** We have reworded as "where $Sc$ and $Sc_{20}$ are the Schmidt number at SST and 20ºC,
respectively".

Line 162-162: "some models overestimated the extent of suboxic water, which might be due to
over-estimated productivity in the euphotic zone" This conclusion does not seem in agreement
with results of Exp1 and Exp2 that show that a decrease in respiration does not allow the
representation of asymmetric OMZ

**Response:** That sentence has been removed because we have rewritten that paragraph due to the
changes in sensitivity experiments and model validations.

Line 225: I find that the use of smaller size is confusing. I guess that the authors mean smaller amount. (and not particles size since the DOM is dissoved).

**Response:** Yes, we have changed to "a smaller amount of DOM".

Line 245: the authors mention that the asymmetric features in many physical and biological fields in the Tropical Pacific are largely associated with asymmetries in water mass exchange between the equatorial and off-equator Pacific Ocean. However, here they use an enhanced vertical diffusion to create this asymmetry and this is not clear how this parameterization can mimic asymmetry water masse exchanges with the regional outside the Pacific.

**Response:** Thank you for your constructive comment. We have re-assessed the model experiments, and made some changes in the sensitivity study which includes new simulations with smaller parameters for background diffusion.

Section 3 (very broadly) describes the results of the experiments but is placed outside the results section.

**Response:** We have changed this section as Results section.

Figure 5: I would say sensitivity experiments rather than sensitive experiments.

**Response:** Corrected.

Table 3: please correct Pm is a diffusion coefficient and has to be in m2/sec and not /m2/sec.

**Response:** Corrected. In order to compare with others' results, we use $cm^2/s$ rather than $m^2/s$.

**Bibliography**

Duteil, O. and Oschlies, A.: Sensitivity of simulated extent and future evolution of marine suboxia to mixing intensity, Geophysical Research Letters, 38, 2011.

Wang, X. J., Behrenfeld, M., Le Borgne, R., Murtugudde, R., and Boss, E.: Regulation of phytoplankton carbon to chlorophyll ratio by light, nutrients and temperature in the Equatorial Pacific Ocean: a basin-scale model, Biogeosciences, 6, 391-404, 2009a.

Wang, X. J., Le Borgne, R., and Murtugudde, R.: Nitrogen uptake and regeneration pathways in the equatorial Pacific: a basin scale modeling study, Biogeosciences, 6, 2647-2660, 2009b.

Wang, X. J., Murtugudde, R., Hackert, E., Wang, J., and Beauchamp, J.: Seasonal to decadal variations of sea surface pCO(2) and sea-air CO2 flux in the equatorial oceans over 1984-2013: A basin-scale comparison of the Pacific and Atlantic Oceans, Global Biogeochemical Cycles, 29, 597-609, 2015.

Wang, X. J., MurtuguddeA, R., Hackert, E., and Maranon, E.: Phytoplankton carbon and chlorophyll distributions in the equatorial Pacific and Atlantic: A basin-scale comparative study, Journal of Marine Systems, 109, 138-148, 2013.

Zhang, R.-H.: A modulating effect of Tropical Instability Wave (TIW)-induced surface wind feedback in a hybrid coupled model of the tropical Pacific, Journal of Geophysical Research: Oceans, 121, 7326-7353, 2016.

Zhang, R.-H. and Busalacchi, A. J.: Rectified effects of tropical instability wave (TIW)-induced atmospheric wind feedback in the tropical Pacific, Geophysical Research Letters, 35, 2008.

---

## Author Comment (AC2) · 30 Nov 2020

**Response to Anonymous Referee #2**

Subject
Wang and co-authors investigate the impacts of physical supply and biological consumption of dissolved oxygen (DO) on the dynamics and asymmetry of the OMZs in the tropical Pacific. They perform 4 numerical experiments to evaluate the sensibility of the mid-depth oxygen concentration to these aspects in their model, OGCM-DMEC v1.0. The physical supply is evaluated through the background diffusion parameterization (that the authors test by changing a partial mixing parameter) and the effects of biological consumption on oxygen are tested by changing the C:O utilization ratio. The final aim is to advance their model capacity to simulate the oceanic oxygen cycle, and to explore the mechanisms driving the asymmetric OMZs in the tropical Pacific (introduction, l.67-68).

Relevance of the subject
To understand the physical and biological processes responsible for the asymmetry of tropical Pacific OMZs is a topic of great interest for climate modelers, which has currently not been solved.

General comments
However, in its present form, the conclusions of this study bring no new clues of understanding, and do not explore any mechanisms. The authors conclude that both physical supply and biological consumption impact the OMZs extend and vertical structure, which, according to them, has been the subject of numerous previous papers (see l.188-190 or l.219-220).

**Response:** Thank you for the constructive comments. We have made major revisions, including some new experiments and analyses, and rewriting of some sections (e.g., model description, sensitivity experiment, model validation, results and discussions).

While it is a promising approach to explore the DO budget term by term, I recommend to enlarge the analyses to other variables (by characterizing the tropical ocean dynamics with vertical sections of horizontal currents for example, and by giving insights of plankton and nutrients mean-state and variability) in order to explore the mechanisms at play when increasing the background diffusion or decreasing the biological consumption.

**Response:** Thank you for the constructive comments. This basin-scale model was developed to study the upper ocean dynamics for the tropical Pacific, which includes the spatial and temporal variabilities of physical and biogeochemical fields. We have analyzed/validated many physical and biogeochemical variables in our previous studies, e.g., SST (Wang et al., 2006), chlorophyll (Wang et al., 2009a; Wang et al., 2013), nitrogen cycle (Wang et al., 2009b) and carbon cycle (Wang et al., 2015). In this paper, we have added the comparisons of DON and oxygen consumption, which have a direct link to the mid-depth DO dynamics.

Besides I have some reservations about the use of a basin-scale model of the Pacific limited at 20 S and 20 N to study the Pacific OMZs. It seems not very appropriated to model OMZ borders, as these latter are found far north of 20 N. If the aim of the study is to investigate the importance of

DO physical supply, one may not ignore the ventilation processes at play in the OMZ borders
(Bettencourt et al., 2015). And even in a case of tropical study (as reflected by the analyses
restricted to 15 S-15 N), one may not ignore the critical representation of the equatorial
undercurrent (EUC) to model the tropical OMZ structure (Busecke et al., 2019). As both
processes are highly resolution dependent (see for example Fig. 16 in Berthet et al., 2019), I am
surprised to find no discussion and no bibliography on the questions of the appropriate model
resolution needed to get a realistic OMZ structure.

**Response:** Thank you for your constructive comments. This basin-scale model was developed to
study the upper ocean dynamics for the tropical Pacific, with a domain of 30°S-30°N. We have
changed to "sponge area" to 25°-30°, and re-done all model simulations and reproduced all
figures covering 25°S-25°N. We have added some discussion regarding the impacts of horizontal
resolution of the model on the OMZ structure.
Results
Model description: The parameterization of the oxygen cycle needs to be described with more
details. It would help the reader to analyze the results.

**Response:** Thank you for the suggestion. We have added more details for model description,
including model equations and parameters (as appendix A and B).
Validation: In its present form, the model validation may be completed by showing physical
currents and temperature/salinity mean state and variability (at the surface and with a vertical
section along latitudes), OMZ inter-annual variability, ventilation at the OMZ boundaries (as
mesoscale activity has been shown to shape the OMZ) …

**Response:** We agree with that more model validation should be carried out. Since our previous
studies have reported the evaluations of physical (including mesoscale and sub-mesoscale
structures) and many biogeochemical fields (see responses above), this paper mainly reports the
calibration and validation of oxygen related fields (e.g., consumption and DON). We have
analyzed the temporal variability of OMZ in another paper.
As stated l.70, the OGCM-DMEC V1.0 has shown a good model-data agreement in the carbon
cycle for the tropical Pacific Ocean (Wang et al., 2015). This is a good point if the model was
validated on carbon cycle, but the paper needs a true validation on oxygen.

**Response:** We have added more model-data comparisons, using cruises' data for the distribution
of DO.
Specific comments
l.14: 'DO' is used in the abstract, but not defined

**Response:** We have defined DO.
l.53: I would recommend to add the following study to justify that circulation play a dominant
role in regulating the dynamics of tropical OMZs: Busecke, J. J. M., Resplandy, L., & Dunne, J. P.

P. (2019). The Equatorial Undercurrent and the oxygen minimum zone in the Pacific.
Geophysical Research Letters, 46. https://doi.org/10.1029/2019GL082692

**Response:** We have added this reference in the introduction section.

l.54-57: And what about the horizontal resolution of the model ? Using an ESM with a
high-resolution ocean (1/10 ), Busecke et al. (2019) show that a realistic representation of the
Equatorial Undercurrent (EUC) dynamics is crucial to represent the upper OMZ structure and its
temporal variability. They demonstrate that coarser ESMs commonly misrepresent the EUC,
leading to an unrealistic "tilt" of the OMZ (e.g., shallowing toward the east) and an exaggerated
sensitivity to EUC changes overwhelming other important processes like diffusion and biology.
This last aspect would be interesting for your discussion.

**Response:** Thank you for your constructive comments. We have added some discussion
regarding the impacts of model's horizontal resolution on the OMZ structure.

l.61: "A fully coupled basin-scale physical-biogeochemical model (OGCM-DMEC V1.0) was
developed for the tropical Pacific (Wang et al., 2008; Wang et al., 2015; Wang et al., 2009)." –>
are you using a regional configuration centered on the Pacific ocean ? Or is it a global model ?

**Response:** Our model is a regional model, with a domain of 30°S-30°N.

l.78: "The model domain is between 30 S and 30 N" –> thus it is not "global". This has to be
clarified, as OGCM generally means ocean GLOBAL circulation model.
Moreover if your domain extends between 30 S-30 N: why did you crop your horizontal maps at
15 N while Fig. 2b clearly not catch OMZ northern border between 200-600m (which seems far
north) ? I would suggest to enlarge the northern border up to 20 N (at least, as your sponge layers
are in the 20 -30 bands). "and zonal resolution is 1." –> have you checked how your EUC
behaves ?

**Response:** Our model is a basin-scale OGCM, and we have used such name/definition in many
our previous publications. Others have also used "OGCM" for a regional ocean model (e.g.,
Sofianos and Johns, 2003).

We have changed to "sponge area" to 25°-30°, and re-done all model simulations and reproduced
all figures covering 25°S-25°N. We believe that our model does a good job in simulating physical
fields including EUC because we have validated many physical and biogeochemical variables in
our previous studies, e.g., TIW (Zhang, 2016; Zhang and Busalacchi, 2008), SST (Wang et al.,
2006), nitrogen cycle (Wang et al., 2009b) and carbon cycle (Wang et al., 2015).

l.86-87: precipitation (gpcp) and wind stress (NCEP) forcings are not consistent ?

**Response:** We have changed to "Precipitation is from ftp://ftp.cdc.noaa.gov/Datasets/gpcp. Wind
stresses are from the National Centers for Environmental Prediction (NCEP) reanalysis (Kalnay
et al., 1996)".

l.90: "an interannual simulation for the period of 1978-2000, and analyze model output for the
period of 1981-2000." –> could you give some insights about the interannual behaviour of your
OMZs ?

**Response:** We have analyzed the temporal variability of OMZ in another paper. This paper has a
focus on the spatial pattern, with a particular interest in the asymmetry of OMZ.

l.96: DON is not defined

**Response:** We have defined DON.

l. 107-109: please clarify your computation of the vertical mixing term: "the vertical mixing term
that is calculated by three subroutines (Chen et al., 1994)." –> I guess that to be splitted in 3
subroutines is not the main characteristic of the hybrid scheme of Chen et al. (1994). It would be
interesting to mention why you add this mixing scheme in your model from a physical point of
view. Following the abstract of Chen et al. (1994), this hybrid vertical mixing scheme "helps to
produce more realistic velocity profiles in the eastern and central equatorial Pacific. This is
mainly due to the improved parameterization of interior mixing related to the large shears of the
Equatorial Undercurrent", which seems to me an important aspect when modelling the OMZ. Or
it would be important to tell the reader (still from their abstract) that this scheme "is capable of
simulating the three major mechanisms of vertical turbulent mixing in the upper ocean, that is,
wind stirring, shear instability, and convective overturning." l. 137: as you aim to determine the
respective roles of physics and biogeochemistry in the oxygen biases, it would be helpful to have
some basic validations on horizontal / vertical circulations (for example, a vertical section of
zonal jets along the latitudes) and nutrients affecting the oxygen budget in your model (phyto-
and zoo-plankton, detritus, DON, ammonium, nitrate).

**Response:** Thank you for your constructive comments. You are correct about the "3 subroutines"
and Chen mixing scheme. We have made a correction, i.e., delete "(Chen et al., 1994)" in that
sentence. We have added some more information about the Chen scheme, as suggested.

l.142-145: Regarding your sensitivity experiments, it would be helpful to clarify how the initial
DON remineralization constant and O:C utilization ratio were determined.

**Response:** Thank you for your suggestion. We have added more information about the model and
parameters, e.g., "The equations for biogeochemical processes and model parameters are
described in Appendix A and B. There have been changes in some parameters comparing with
those in Wang et al. (2008), which were based on our model calibration and validation for
chlorophyll (Wang et al., 2009a, Wang et al., 2013), nitrogen cycle (Wang et al., 2009b) and
carbon cycle (Wang et al., 2015)". Note: we have deleted the modified O:C simulation because
there was no good reason to change the O:C ratio.

Moreover, are you increasing the oxygen supply through mixing only in the OMZ region or in the
whole Pacific basin? Could you justify your choices ? Could you elaborate on your "variable
Pm" ? How does it vary?

**Response:** All the experiments with enhanced mixing are in the whole basin. We have added
more information to justify our choices (see responses above). We have deleted the "variable Pm"
experiment.
l.179-180: "We first compare the distribution of DO over 300-500 m between reference run and
Exp3. The reference run produces much large volume of suboxic waters (<20 mmol m-3) in both
the ETNP and ETSP where the two OMZs are merged (Figure 6a)." The reader would appreciate
if the oxygen average for your "ref" experiment in Fig. 6 may be comparable with observations:
Fig. 2 (right column) shows the 200-600 m mean, and Fig. 6 the 300-500 m mean. These 2
averaged layers (200-600 vs 300-500 m) are quite different in terms of volume of equatorial
suboxic waters, so, please, could you add a 3rd column in Fig. 2 with the 300-500 m mean in
WOA ?

**Response:** We have made major revisions, with all figures showing the comparison/analyses over
200-400 m, 400-700 m, and 700-1000 m.
l. 181: "Exp3 performs well in reproducing the sizes and locations of two asymmetric OMZs" –>
the use of quantitative metrics (OMZ volume, maximal horizontal extent) would reinforce this
conclusion.

**Response:** Thanks for the suggestion. We have added a new table to show the quantitative
metrics for model-data comparison (see below).

**Table 3.** Comparisons of OMZ volume ($10^{15}$ m$^3$) between WOA2013 and model experiments

| Regions | Waters | WOA2013 | Reference | CD0.5 | CD0.5 PM0.1 | CD0.5 PM0.3 | CD0.5 PM0.5 | CD0.5 PM1.0 |
|---------|--------|---------|-----------|-------|-------------|-------------|-------------|-------------|
| North Pacific | Suboxic | 5.97 | 10.47 | 8.87 | 8.29 | 7.36 | 6.61 | 5.23 |
| | Hypoxic | 19.98 | 21.21 | 20.48 | 20.35 | 20.01 | 19.62 | 18.74 |
| South Pacific | Suboxic | 1.43 | 3.49 | 2.42 | 2.20 | 1.85 | 1.56 | 0.93 |
| | Hypoxic | 7.12 | 9.90 | 8.73 | 8.35 | 7.70 | 7.13 | 5.96 |

Suboxic: DO <20 mmol m$^{-3}$; Hypoxic: DO <60 mmol m$^{-3}$.
l.195: regarding the small decrease you detect in the ETNP-OMZ in exp3 (Fig. 7c): what do you
obtain with exp4 ? Is this decrease linked with coastal processes ? If yes, how ?

**Response:** The small decrease of physical supply in the ETNP-OMZ is detected in all the
experiments with enhanced background diffusion. We think that this decrease is not linked with
coastal processes, but is due to the redistribution of DON, which causes non-uniform changes in
consumption over depth, and thus alters the vertical gradients of DO (see figure below).

[Figure]

**Figure 10.** Changes due to enhanced background diffusion (CD0.5PM0.5 minus CD0.5) for (a) DO, (b)
physical supply, (c) biological consumption, and (d) DON. Red lines (ETNP: 165°W-90°W, 5°N-20°N)
and blue lines (ETSP: 110°W-80°W, 15°S-5°S).

Figures

Figure 1: legend of Ps, PL, Zs, ZL, Ds, DL is missing.

**Response:** We have redrawn the ecosystem diagram (see figure below).

[Figure]

**Figure 1.** Flow diagram of ecosystem model. Red, green, blue, yellow and brown lines and arrows denote
fluxes originating from inorganic forms, phytoplankton, zooplankton, DON and detritus, respectively.

Figure 2: it seems weird to me to study the Pacific OMZ but to not catch its spatial extend
entirely: why don't you extend your simulated regions at least to 25 S and 25 N (shifting your
sponge layers between 30 and 35 for example), and to the coasts of America (70W to get both
northern and southern parts of the Pacific OMZ) ?

**Response:** Thank you for the suggestion. We have extended simulated region to 25°S and 25°N,
and to the coasts of America (see figure below).

[Figure]

**Figure 2.** Comparisons of DO concentration between WOA2013 (left panel) and reference run over
1981-2000 (right panel).

Figures 3 (and 10): as the paper focus on the asymmetry between the northern and southern part
of the Pacific OMZ, and as its aim is to show how they differ, the meridional means between
10S-15N seem not appropriate. I would recommend to split the analyse in two, one for each OMZ
(south and north). As it is, Fig. 3 does not allow to properly evaluate how the model reproduces
the vertical structure of the OMZ against observations. Same comment for Fig. 10 (left column),
and this analyse does not allow to investigate any mechanisms.

**Response:** Thank you for your suggestion. We have made major revisions, which include the
deletion of previous Figure 3 because we think that Figure 4 (see below) is sufficient to show the
asymmetric OMZs. Regarding Figure 10 (now Figure 9), we have split the zonal distribution into
two as suggest (see below).

[Figure]

**Figure 4.** Observed and simulated DO from model experiments over 130°W-90°W. (a) WOA2013, (b) reference run, (c) CD0.5, (d) CD0.5PM0.1, (e) CD0.5PM0.5, and (f) CD0.5PM1.0 over 1981-2000.

[Figure]

**Figure 9.** Distribution of DON over 130°W-90°W (left), 5°N-20°N (middle), and 15°S-5°S (right) from (a-c) CD0.5, (d-f) CD0.5PM0.5, and (e-f) differences (CD0.5PM0.5 minus CD0.5).

**Bibliography**

Sofianos, S. S. and Johns, W. E.: An Oceanic General Circulation Model (OGCM) investigation of the Red Sea circulation: 2. Three-dimensional circulation in the Red Sea, J Geophys Res-Oceans, 108, 2003.

Wang, X. J., Behrenfeld, M., Le Borgne, R., Murtugudde, R., and Boss, E.: Regulation of phytoplankton carbon to chlorophyll ratio by light, nutrients and temperature in the Equatorial Pacific Ocean: a basin-scale model, Biogeosciences, 6, 391-404, 2009a.

Wang, X. J., Christian, J. R., Murtugudde, R., and Busalacchi, A. J.: Spatial and temporal variability of the surface water pCO(2) and air-sea CO2 flux in the equatorial Pacific during 1980-2003: a basin-scale carbon cycle model, Journal of Geophysical Research, 111, C07S04, doi:10.1029/2005JC002972, 2006.

Wang, X. J., Le Borgne, R., and Murtugudde, R.: Nitrogen uptake and regeneration pathways in the equatorial Pacific: a basin scale modeling study, Biogeosciences, 6, 2647-2660, 2009b.

Wang, X. J., Murtugudde, R., Hackert, E., Wang, J., and Beauchamp, J.: Seasonal to decadal variations of sea surface pCO(2) and sea-air CO2 flux in the equatorial oceans over 1984-2013: A basin-scale comparison of the Pacific and Atlantic Oceans, Global Biogeochemical Cycles, 29, 597-609, 2015.

Wang, X. J., MurtuguddeA, R., Hackert, E., and Maranon, E.: Phytoplankton carbon and chlorophyll distributions in the equatorial Pacific and Atlantic: A basin-scale comparative study, Journal of Marine Systems, 109, 138-148, 2013.

Zhang, R.-H.: A modulating effect of Tropical Instability Wave (TIW)-induced surface wind feedback in a hybrid coupled model of the tropical Pacific, Journal of Geophysical Research: Oceans, 121, 7326-7353, 2016.

Zhang, R.-H. and Busalacchi, A. J.: Rectified effects of tropical instability wave (TIW)-induced atmospheric wind feedback in the tropical Pacific, Geophysical Research Letters, 35, 2008.